# The mitochondrial deoxyguanosine kinase is required for cancer cell stemness in lung adenocarcinoma

Shengchen Lin[1,†], Chongbiao Huang[2,†], Jianwei Sun[1,3,4], Oana Bollt[5], Xiuchao Wang[1,2], Eric Martine[1], Jiaxin Kang[1,4], Matthew D Taylor[5], Bin Fang[6], Pankaj K Singh[7], John Koomen[6], Jihui Hao[2,*] (iD) & Shengyu Yang[1,8,**] (iD)

## Abstract

The mitochondrial deoxynucleotide triphosphate (dNTP) is maintained by the mitochondrial deoxynucleoside salvage pathway and dedicated for the mtDNA homeostasis, and the mitochondrial deoxyguanosine kinase (DGUOK) is a rate-limiting enzyme in this pathway. Here, we investigated the role of the DGUOK in the self-renewal of lung cancer stem-like cells (CSC). Our data support that DGUOK overexpression strongly correlates with cancer progression and patient survival. The depletion of DGUOK robustly inhibited lung adenocarcinoma tumor growth, metastasis, and CSC self-renewal. Mechanistically, DGUOK is required for the biogenesis of respiratory complex I and mitochondrial OXPHOS, which in turn regulates CSC self-renewal through AMPK-YAP1 signaling. The restoration of mitochondrial OXPHOS in DGUOK KO lung cancer cells using NDI1 was able to prevent AMPK-mediated phosphorylation of YAP and to rescue CSC stemness. Genetic targeting of DGUOK using doxycycline-inducible CRISPR/Cas9 was able to markedly induce tumor regression. Our findings reveal a novel role for mitochondrial dNTP metabolism in lung cancer tumor growth and progression, and implicate that the mitochondrial deoxynucleotide salvage pathway could be potentially targeted to prevent CSC-mediated therapy resistance and metastatic recurrence.

**Keywords** cancer stem cell; DGUOK; lung cancer; metastasis; mitochondria
**Subject Categories** Cancer; Stem Cells & Regenerative Medicine; Respiratory System

## Introduction

Lung cancer is the leading cause of cancer-related death in the United States. Non-small-cell lung cancer (NSCLC) accounts for about 85% of all lung cancers, and lung adenocarcinoma is the most common NSCLC (Nesbitt *et al*, 1995; Siegel *et al*, 2015). Most lung cancer patients eventually succumb to local and metastatic recurrence, including many patients who underwent complete removal of primary tumor and displayed no detectable metastasis at the time of surgery (Nesbitt *et al*, 1995; Mountain, 1997; D'Amico *et al*, 1999). It is believed that a subset of slow-cycling cells with stem cell-like properties termed cancer stem-like cells (CSC) are responsible for tumor initiation and local or metastatic recurrence (Pardal *et al*, 2003; Li *et al*, 2007; Chaffer & Weinberg, 2011). CSC are resistant to conventional chemo- and radiotherapies and are able to remain dormant for many years before triggering metastatic recurrence (Chaffer & Weinberg, 2011). Conventional cancer therapy aims at eliminating proliferating cancer cells. However, development of a strategy to eradicate disseminated CSC and to prevent metastatic recurrence is still a major challenge.

There are two distinct pools of deoxynucleotide triphosphate (dNTP) in eukaryotic cells (Kohnken *et al*, 2015). The *de novo* synthesis of dNTP, in the cytosol, is coordinated with the cell cycle and peaks at the S-phase to supply deoxynucleotides for the replication of genomic DNA (Kohnken *et al*, 2015). In contrast, the mitochondrial pool of dNTP is maintained throughout the cell cycle by the mitochondrial deoxynucleoside salvage pathway and is crucial for mtDNA replication (Wang & Eriksson, 2003; Sandrini & Piskur, 2005; Bulst *et al*, 2009). The *de novo* synthesis of cytosolic dNTP by the ribonucleotide reductase (RNR) has been extensively studied in cancer and is

1 Department of Cellular and Molecular Physiology, The Pennsylvania State University College of Medicine, Hershey, PA, USA
2 Key Laboratory of Cancer Prevention and Therapy, Department of Pancreatic Cancer, Tianjin Medical University Cancer Institute and Hospital, National Clinical Research Center for Cancer, Tianjin, China
3 State Key Laboratory of Natural Resource Conservation and Utilization in Yunnan and Center for Life Science, School of Life Sciences, Yunnan University, Kunming, China
4 South China Agricultural University, Guangzhou, China
5 Department of Surgery, The Pennsylvania State University College of Medicine, Hershey, PA, USA
6 Department of Molecular Oncology, Proteomics & Metabolomics Core, H. Lee Moffitt Cancer Center, Tampa, FL, USA
7 Department of Pathology and Microbiology, Eppley Institute for Research in Cancer and Allied Diseases, University of Nebraska Medical Center, Omaha, NE, USA
8 Department of Tumor Biology, H. Lee Moffitt Cancer Center, Tampa, FL, USA
  *Corresponding author. Tel: +86 22 23340123 extension 3070; E-mail: haojihui@tjmuch.com
  **Corresponding author. Tel: +1 717 531 1721; E-mail: sxy99@psu.edu
  †These authors contributed equally to this work

believed to be one of the most frequently dysregulated pathways during tumorigenesis (Mathews, 2015). Many FDA-approved anti-cancer agents such as 5-fluorouracil, gemcitabine, and 6-mercapto-purine are believed to act at least in part by disrupting metabolism of the cytosolic deoxynucleotide (Mathews, 2015). However, little is known about the role of mitochondrial dNTP metabolism in cancer.

Mitochondria are the powerhouses of the cell crucial for both anabolic and catabolic metabolism. The mitochondrial oxidative phosphorylation (OXPHOS) is crucial for the self-renewal of CSC in lung cancer, glioblastoma, and leukemia (Ye *et al*, 2011; Janiszewska *et al*, 2012; Sancho *et al*, 2015). The human mitochondrial DNA (mtDNA) encodes 13 mitochondrial respiratory chain proteins essential for OXPHOS. The replication of mtDNA is essential for the cell to replenish damaged mitochondria and to maintain mitochondrial functionality. The dNTP pool used for mtDNA replication could be provided by the *de novo* synthesis pathway or the salvage pathway (Franco *et al*, 2007; Kohnken *et al*, 2015). In quiescent and slow-cycling cells (such as CSC), the *de novo* dNTP synthesis is suppressed, and the replication of mtDNA depends on the mitochondrial deoxynucleoside salvage pathway (Franco *et al*, 2007). Thymidine kinase 2 (TK2) and deoxy-guanosine kinase (DGUOK) are two mitochondrial deoxynucleoside kinases responsible for the salvage of pyrimidine and purine deoxynucleoside, respectively, in mitochondria. TK2 and DGUOK are crucial for the phosphorylation of anti-viral and anti-leukemic nucleoside analog pro-drugs such as forodesine and nelarabine. TK2 and DGUOK play an important role in the conversion of these pro-drugs to active compounds (Rodriguez *et al*, 2002). However, the role of the mito-chondrial deoxynucleoside salvage pathway in tumorigenesis and tumor progression has not been previously reported.

Here, we examined the mitochondrial deoxynucleoside salvage pathway as a potential target in lung cancer CSC. Our data indicate that DGUOK is overexpressed in lung adenocarcinoma patients and the expression levels of DGUOK, but not TK2, strongly correlate with the survival of lung adenocarcinoma patients. Our data further support that DGUOK targeting inhibits the self-renewal of lung cancer CSC, mitochondrial respiration, and AMPK-YAP signaling. Our proof-of-concept study indicates that the mitochondrial deoxy-guanosine kinase could be targeted to inhibit mitochondrial metabolism and CSC self-renewal in lung adenocarcinoma. Pharmacological inhibitors targeting DGUOK could be useful in the treatment of lung cancer and the prevention of lung cancer recurrence.

## Results

### DGUOK overexpression is required for lung adenocarcinoma tumor growth and metastasis

To understand the role of mitochondrial deoxynucleoside salvage pathway in lung cancer progression, we evaluated the correlation between *TK2* and *DGUOK* mRNA transcripts with the survival of lung cancer patients in a previously published meta-analysis dataset (www.kmplot.com; Gyorffy *et al*, 2010). Interestingly, the expression levels of *DGUOK*, but not *TK2*, strongly correlated with the survival of lung adenocarcinoma patients (Figs 1A and EV1A). To further determine the role of DGUOK in lung adenocarcinoma progression, we used immunohistochemistry (IHC) to evaluate DGUOK expression levels in 82 pairs of lung adenocarcinoma specimens and para-tumor

lung tissues (Fig 1B). DGUOK staining in lung adenocarcinoma tissues appeared as punctate or diffuse cytosolic staining in lung adenocarcinoma specimens and formalin-fixed paraffin-embedded H1650 cells (Fig EV1B and C), which is consistent with its mitochondrial localization in the cell. The staining was not detected in DGUOK KO H1650 cells, confirming the specificity of this antibody (Fig EV1B). DGUOK expression was highly elevated in lung adenocarcinoma when compared to paired adjacent lung tissues, indicating upregulation of DGUOK in lung cancer (Fig 1B). To further determine the role of DGUOK in lung cancer progression, we investigated the correlation between DGUOK levels and clinicopathological parameters in a lung adenocarcinoma tissue microarray consisting of 113 tumor samples (Fig 1C). The expression levels of DGUOK were robustly correlated with tumor size, nodal involvement, and TNM staging but not with the age, gender, or smoking status of patients (Table EV1). Patients with high DGUOK expression have shorter overall survival when compared to those with low or no DGUOK expression (Fig 1C and D). Multivariable analysis using Cox's proportional hazards model showed that DGUOK is an independent prognostic factor among lung adenocarcinoma patients (Table 1), implicating a critical role of DGUOK in lung adenocarcinoma progression.

To understand the functional role of DGUOK in lung adenocarcinoma, we employed CRISPR/Cas9 to KO DGUOK expression in lung adenocarcinoma cells (Fig EV1D). DGUOK depletion with two separate sgRNA dramatically inhibited soft-agar colony formation in LLC, H1650, and A549 cells (Fig 1E), although it only modestly inhibited cell proliferation under adherent conditions (Fig EV1E). When implanted subcutaneously, the tumor growth of DGUOK KO H1650 cell was much slower when compared to the control group (Fig 1F). To determine the role of DGUOK in metastatic colonization, luciferase-labeled control or DGUOK KO H292 cells were injected via tail vein into nude mice, and the development of lung metastases was monitored through bioluminescence imaging. As shown in Fig 1G and H, DGUOK depletion in H292 cells inhibited lung colonization by over 90%. DGUOK KO also inhibited lung cancer cell migration and invasion without affecting the expression of epithelial–mesenchymal transition markers (Appendix Fig S1A–C). We further investigated the role of DGUOK in lung cancer tumor growth and metastasis in an orthotopic model. Luciferase-labeled control or DGUOK KO H1650 cells were orthotopically injected into the left lung of nude mice using an established protocol (Onn *et al*, 2003). We confirmed, in a pilot experiment, that using this technique, lung cancer cells were reliably injected into the left lung of the mice (Appendix Fig S1D–I). The growth of orthotopic H1650 xenograft was monitored through biolu-minescence imaging (Fig EV1F and G). At 26 day after implantation, mice were euthanized and the growth of primary tumor (left lung) and local metastasis (right lung) was monitored by *ex vivo* biolumi-nescence imaging (Fig 1I). The depletion of DGUOK inhibited the growth of orthotopic primary tumor (left lung) by 75% and the development of local metastases (right lung) by 91% (Fig 1J). Taken together, our data indicate that DGUOK overexpression in lung adeno-carcinoma is essential for both tumor growth and metastasis.

### DGUOK is required for cancer cell stemness in lung adenocarcinoma

Mitochondrial respiration has been recently implicated in maintaining cancer cell stemness (Sancho *et al*, 2016; Loureiro *et al*, 2017).

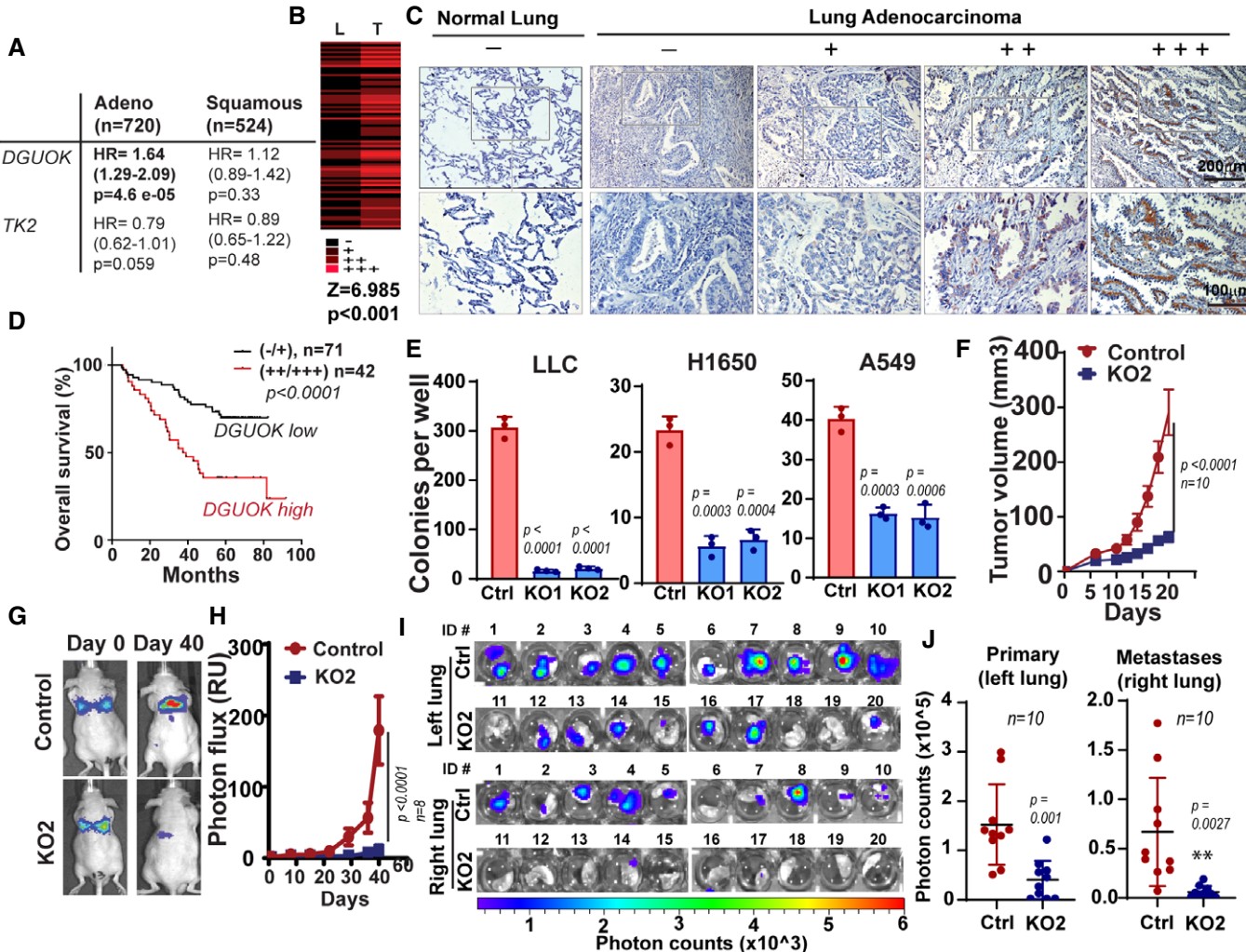

**Figure 1. DGUOK overexpression in lung adenocarcinoma is required for tumor growth and metastasis.**

A    Kaplan–Meier analysis showing the correlation between *DGUOK* and *TK2* expression levels and patient overall survival in lung adenocarcinoma patients and lung squamous cell carcinoma in a meta-analysis dataset (kmplot.com). HR, hazard ratio.

B    DGUOK expression in lung adenocarcinoma (T) and paired para-tumor lung tissues (L), as determined by IHC. *P* value was determined by two-tailed Wilcoxon signed-rank tests.

C    Representative images showing DGUOK IHC staining in normal lung and lung adenocarcinoma in a tissue microarray.

D    The correlation between DGUOK expression levels and overall survival rate in lung adenocarcinoma patients. $P < 0.0001$, log-rank test.

E    The effects of DGUOK knockout (KO1 and KO2) by two independent sgRNA on soft-agar colony formation in lung adenocarcinoma cells. Data are shown as mean $\pm$ SD. $n = 3$ biological replicates per group. Experiments were repeated with three biological replicates with similar results.

F    The effects of DGUOK depletion on H1650 xenograft s.c tumor growth. Data are shown as mean $\pm$ SD ($n = 10$ mice per group). $5 \times 10^6$ cells were inoculated.

G, H    Representative bioluminescence imaging (G) and quantitation of bioluminescence signals (H) showing the effects of DGUOK depletion on lung colonization by tail vein-injected H292 cells. Data are shown as mean $\pm$ SD ($n = 8$ mice per group). $5 \times 10^6$ cells were inoculated to each mouse.

I    *Ex vivo* bioluminescence imaging of extracted lungs from nude mice receiving orthotopic implantation of $1 \times 10^6$ H1650 cells to the left lung.

J    Quantitation of bioluminescence imaging data from primary orthotopic lung tumor (left lung) and local metastasis (right lung).

Data information: *P* values were determined by two-tailed, two-sample Student's *t*-test.

Since the homeostasis of mtDNA in slow-cycling cells (such as CSC) depends on the mitochondrial deoxynucleoside salvage pathway, we hypothesized that DGUOK depletion might inhibit lung cancer tumor growth and metastasis through suppressing cancer cell stemness. To investigate this hypothesis, we determined whether DGUOK depletion affected the proportion of CSC in lung cancer cell lines using several markers, including aldehyde dehydrogenase (ALDH) activity (Sullivan *et al*, 2010; Shao *et al*, 2014), CD166, CD49, and the side populations (Ho *et al*, 2007; Zhang *et al*, 2012, 2017; Bora-Singhal *et al*, 2015). We first determined the effects of DGUOK depletion on the proportion of ALDH⁺ population in lung adenocarcinoma cells. H1650 and LLC cells were loaded with ALDE-FLUOR, a fluorescent substrate for aldehyde dehydrogenase, and effects of DGUOK depletion on the proportions of CSC with high ALDH activities were determined by flow cytometric (Appendix Fig S2A). As shown in Figs 2A and EV2A, DGUOK depletion in H1650

**Table 1. Cox's proportional hazards model analysis of prognostic factors in patients with lung adenocarcinoma (LUAD).**

| Variables | Unfavorable/ favorable | Overall survival | |
|---|---|---|---|
| | | HR(95% CI) | P value |
| Univariate analysis | | | |
| Gender | Male/female | 1.262 (0.719–2.215) | 0.418 |
| Age | ≥ 65/65 | 0.952 (0.512–1.773) | 0.877 |
| Smoker | Smoker/non-smoker | 1.261 (0.712–2.235) | 0.427 |
| Tumor size | > 3.5/≤ 3.5 | 1.966 (1.094–3.530) | 0.024* |
| T staging | T2b-T4/T1-T2a | 1.755 (0.990–3.109) | 0.054 |
| pN factor | +/− | 1.716 (0.978–3.009) | 0.060 |
| TNM staging | IIB-III/I-IIA | 1.927 (1.089–3.413) | 0.024* |
| DGUOK level | High/low or absent | 3.072 (1.732–5.449) | 0.0001* |
| Multivariable analysis | | | |
| Tumor size | > 3.5/≤ 3.5 | 1.644 (0.897–3.011) | 0.107 |
| Age | ≥ 65/65 | 0.839 (0.449–1.567) | 0.581 |
| Smoker | Smoker/non-smoker | 1.257 (0.704–2.246) | 0.439 |
| DGUOK level | High/low or absent | 2.737 (1.512–4.953) | 0.001* |

Asterisk (*) indicates P < 0.05.

and LLC cells decreased the ALDH$^+$ population by twofold or fivefold, respectively. Secondly, we used CD166 [another well-established lung CSC marker (Zhang et al, 2012, 2017)] antibody to label control and DGUOK KO H1650 and A549 cells (Appendix Fig S2B). As shown in Fig 2B, DGUOK depletion also significantly decreased the CD166$^+$ population in lung cancer cells, further supporting that DGUOK depletion decreased the CSC population in lung adenocarcinoma. The co-staining of lung cancer cells with CD166 and ALDE-FLUOR indicated that the majority (approximately 78%) of the ALDH$^+$ cells were also CD166 (Fig EV2B). The CD166$^+$/ALDH$^+$ cells decreased from 25.8% in control H1650 cells to 1.21% in DGUOK KO H1650. CD49f has also been reported to label CSC in lung cancer patient specimens when used together with CD166 (Zhang et al, 2017). However, our data indicate that CD49f is highly expressed in almost 100% of lung cancer cell lines (Fig EV2C and D). It has been reported that CSCs in several cancer types (including lung cancer) were able to excrete Hoechst dyes through ABCG2 transporter and thus form distinct Hoechst negative "side populations" (Ho et al, 2007; Bora-Singhal et al, 2015). Our data showed that DGUOK KO robustly decreased the proportion of "side population" CSC in H1650 and A549 (Fig EV2E). In summary, using different assays for lung cancer CSC markers, our results consistently demonstrate a significant reduction in lung cancer CSC population after DGUOK KO.

The ability of individual lung cancer cells to form tumor sphere under non-adherent condition in serum-free medium is considered a surrogate assay for the self-renewal abilities of cancer cell stemness. To further investigate the role of DGUOK in CSC self-renewal, we determine the effects of DGUOK depletion on lung sphere formation in lung cancer cell lines and patient-derived lung adenocarcinoma cells (PDCs). The expression levels of DGUOK were comparable between established lung cancer cell lines (H1650, A549) and PDCs (Fig EV2F). As shown in Figs 2C and D, and EV2G, the depletion of DGUOK dramatically inhibited tumor sphere formation in LLC,

H1650, A549, and three PDCs. Although DGUOK depletion slightly increased cell apoptosis in H1650 cell under adherent conditions (5.8 ± 0.5% in control cells vs. 6.8 ± 0.4% in DGUOK KO cells), there was no difference in apoptosis under sphere-forming non-adherent condition (28.1 ± 2.1% in the control vs. 29.8 ± 2.6% in the KO groups) suggesting that the difference in tumor sphere-forming capacity was not due to anoikis resistance (Fig 2E and F). Another important characteristic of CSC is their ability to transdifferentiate (Lin et al, 2012; Shekhani et al, 2013). When embedded in Matrigel, H1650 and A549 cells were able to form undifferentiated tumor spheres with low or negative CK20 expression (Fig 2G and H). After exposure to serum, the expression levels of CK20 were markedly increased in control tumor spheres (Fig 2G and H). The KO of DGUOK in these two cell lines dramatically inhibited the formation of tumor spheres and the expression of serum-induced CK20, suggesting DGUOK is critical for the transdifferentiation of lung cancer cells. Taken together, our data support a role for DGUOK in lung cancer cell stemness.

## DGUOK is required for mitochondrial OXPHOS in lung adenocarcinoma cells

To understand the mechanism by which DGUOK regulates cancer cell stemness, we used quantitative liquid chromatography–tandem mass spectrometry to investigate the changes in protein expression between controls and DGUOK KO cells. Over 6,000 proteins were detected by LC-MS/MS. The expression levels of 131 proteins were differentially expressed (fold changes more than 2 standard deviation away from the mean) in DGUOK knockout H1650 cells (Table EV2). Among the 17 differentially expressed mitochondrial proteins are two subunits of the respiratory chain complex I (NDUFB10 and NDUFB8) and one subunit of complex IV (COX6A1) (Fig 3A). GSEA (Gene Set Enrichment Analysis) of the TCGA lung adenocarcinoma RNA-seq data (Campbell et al, 2016) also indicate that the expression of oxidative phosphorylation pathway genes is highly enriched in DGUOK high lung adenocarcinoma patients (Fig 3B and C), implicating a role for DGUOK in the regulation of mitochondrial respiration in these patients. To investigate the effects of DGUOK depletion on mitochondrial respiratory complexes, we used Western blotting to determine the levels of complex I and complex IV subunits in control and DGUOK KO H1650 cells. As shown in Fig 3D, the expression levels of several complex I and complex IV proteins (mt-ND1, NDUFB8, NDUFB10, mt-CO2) are remarkably decreased in DGUOK KO cells. The decrease in respiratory complex proteins was further confirmed when a different sgRNA was used to knockout DGUOK (KO2) in lung cancer cells (Fig EV3A). In contrast, the KO of DGUOK in fibroblast cells (NIH3T3) had no effect on the expression of complex I proteins (Fig EV3A). The reduced expression of complex I proteins was not due to global inhibition of mitochondrial biogenesis, since the protein levels of TOM20 (a mitochondrial outer membrane protein) were not affected (Fig 3D). The mitochondrial mass (as determined by MitoTracker Green and NAO staining) in DGUOK KO cells was only decreased by approximately 15% (Fig EV3B), further indicating that DGUOK depletion had limited effect on global mitochondrial biogenesis.

To determine the effect of DGUOK depletion on mitochondrial OXPHOS, we used Mito Stress tests to evaluate the oxygen consumption rate (OCR) of H1650 cells in the presence of

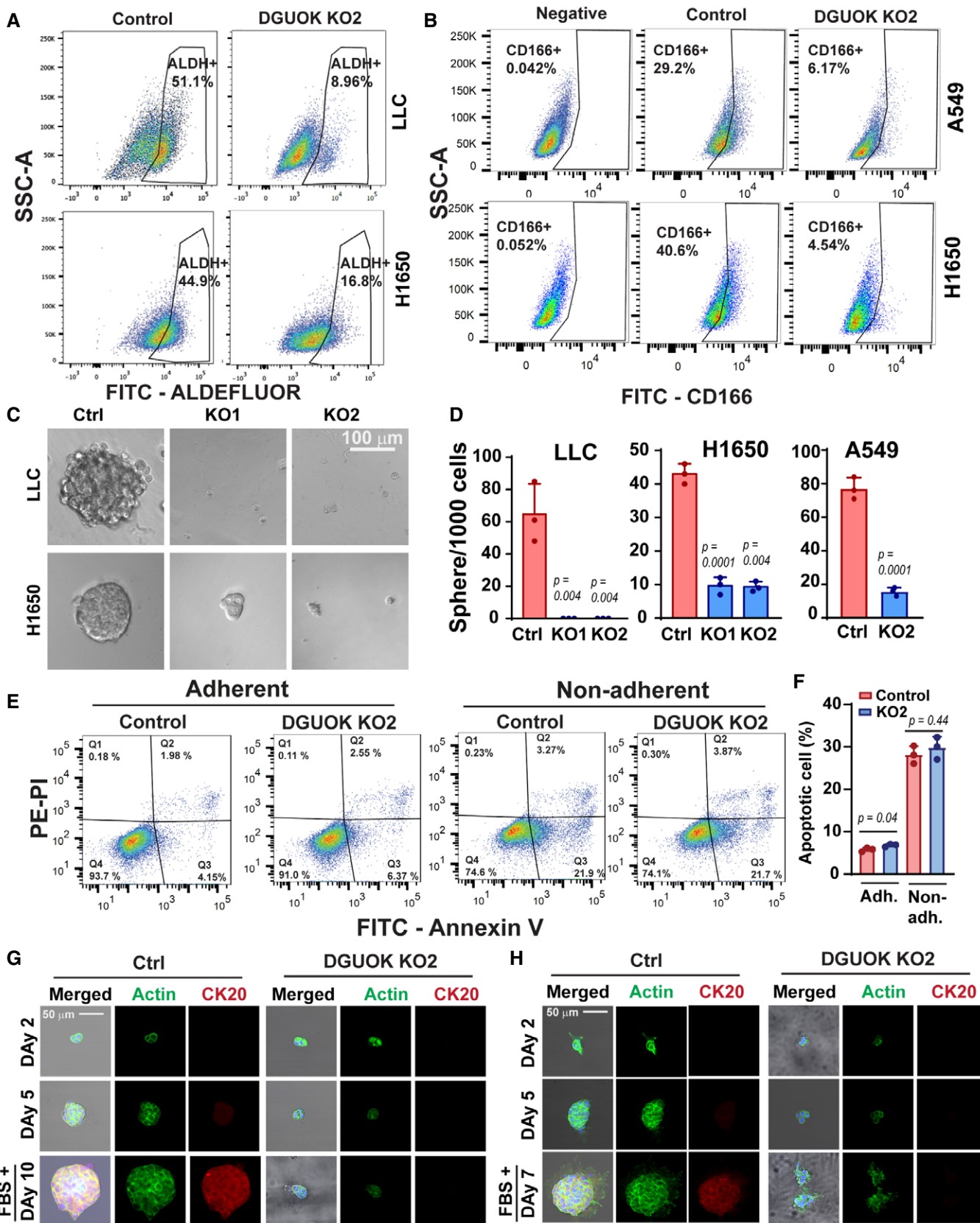

**Figure 2.**

Figure 2.  DGUOK is required for cancer cell stemness in lung adenocarcinoma cells.

A, B    Flow cytometric assay using ALDEFLUOR (A) and anti-CD166 antibody (B) showing the effects of DGUOK depletion on the proportion of ALDH[+] cells (A) and CD166[+]
        cells (B) in lung adenocarcinoma cell lines.
C, D    Representative images and quantitation of the effects of DGUOK depletion on lung tumor sphere formation by lung adenocarcinoma cells.
E       Cell survival assay using annexin V/PI staining and flow cytometric to determine the effects of DGUOK depletion on H1650 cell survival under adherent and non-
        adherent (30 h in lung sphere formation conditions) conditions.
F       Quantitation of apoptotic cells in (E) from three independent biological replicates.
G, H    Representative immunofluorescence staining showing the effects of DGUOK KO on expression of CK20 in Matrigel-embedded H1650 (G) and A549 (H) tumor
        spheres. FBS was added at day 5 to induce CD20 expression.

Data information: Data are shown as mean ± SD. $n = 3$ in (D and F). $P$ values were determined by two-tailed, two-sample Student's $t$-test. Experiments were repeated
with three biological replicates with similar results.

respiratory complex inhibitors. DGUOK depletion inhibited ATP-linked OCR and reserve capacity OCR, suggesting a crucial role for DGUOK in the mitochondrial metabolism in lung cancer cells (Figs 3E and EV3C). Since mtDNA encodes 13 protein subunits essential for the respiratory complex biogenesis, we investigated the effects of DGUOK depletion on mtDNA levels in lung cancer cells. DGUOK knockout reduced the mtDNA levels and levels of mtDNA-encoded RNA by 50–70% (Figs 3F and EV3D). Although DGUOK depletion had no effects on the mRNA levels of nuclear-encoded complex I subunits (Fig EV3D), the protein stability of nuclear-encoded NDUFA9 and NDUFB8 was reduced in DGUOK KO cells (Fig EV3E). It is possible that the decrease in mtDNA-encoded proteins inhibits the assembly of respiratory complex I, which in turn reduces the stability of nuclear-encoded subunits. We also observed 56% reduction in complex I activity in DGUOK KO cells, which was consistent with the marked reduction in complex I protein levels (Fig 3G). Complex IV activities were decreased by 15%, while the activities of complexes II and III were not affected in DGUOK KO cells. It is interesting to note that 7 of the 13 mtDNA-encoded proteins are complex I subunits, which could explain the more severe inhibition of complex I activities in DGUOK KO cells.

There is evidence that disruption of mtDNA homeostasis might either increase or decrease the pool of cytosolic dNTP (Nikkanen et al, 2016). Since DGUOK KO decreased the mtDNA levels in lung adenocarcinoma cells, we examined the effects of DGUOK KO on the levels of cellular dNTP pools. As shown in Fig EV3F, the dNTP levels all decreased by 50–70%, suggesting a role for DGUOK in the regulation of cytosolic dNTP metabolism, presumably through mtDNA homeostasis and mitochondrial metabolism.

To determine whether DGUOK may indeed contribute to the biogenesis of mitochondrial respiratory complex I in cancer patients, we used IHC to determine the expression of NDUFB8 in our lung adenocarcinoma tissue microarray. We chose NDUFB8 because this is one of the most dramatically down-regulated complex I proteins in DGUOK KO lung cancer cells, and also because we were able to obtain an anti-NDUFB8 antibody suitable for IHC staining. As shown in Fig 3H, the expression of NDUFB8 co-localized with DGUOK in lung adenocarcinoma patient specimens. Spearman correlation analysis revealed robust correlation between NDUFB8 expression and DGUOK expression in the cohort of lung adenocarcinoma patients ($r = 0.728$, $n = 113$, $P < 0.0001$; Fig 3I). Taken together, our data support that DGUOK is crucial for mitochondrial OXPHOS in lung adenocarcinoma.

## DGUOK depletion inhibits YAP1 signaling through activation of AMPK

Metabolic reprogramming is crucial for maintaining the stemness of CSC (Folmes et al, 2012; Wu et al, 2015; Sancho et al, 2016). Recently, Hippo pathway transcription factors YAP and TAZ emerged as crucial mediator of the metabolic control of CSC self-renewal (Sorrentino et al, 2014; Enzo et al, 2015; Mo et al, 2015; Santinon et al, 2015, 2016). Interestingly, DGUOK depletion significantly reduced the protein levels of YAP1 (but not that of TAZ) (Fig 4A), inhibited the nuclear translocation of YAP1 (Figs 4B and EV4A) and reduced the mRNA transcript levels of YAP/TAZ target genes CTGF and CYR61 (Fig 4C), indicating inhibition of YAP1 signaling in DGUOK KO cells. The decreases in YAP1 protein levels were also observed in DGUOK KO lung adenocarcinoma PDCs (Fig EV4B). Next, we investigated the mechanism by which DGUOK controls YAP1 expression. It is interesting to note that the levels of pS127-YAP1 (LATS1/2 phosphorylated YAP1) also decreased to an extent proportional to the change in total YAP1 (Fig 4A), suggesting that the inhibition of YAP1 after DGUOK depletion was not due to activation of its canonical upstream kinases, LATS1/2.

The metabolic rewiring of cancer cells modulates the activity of the energy stress sensor AMP-activated kinase (AMPK), which in turn controls YAP/TAZ signaling directly (by direct phosphorylation of YAP) or indirectly (by activating LATS1/2 or AMOTL2) (DeRan et al, 2014; Mo et al, 2015; Wang et al, 2015; Santinon et al, 2016). The DGUOK KO lung cancer cells had increased ADP and decreased ATP levels, which resulted in about a twofold increase in ADP/ATP ratios, suggesting that DGUOK depletion resulted in energy stress in these cells (Fig EV4C). Since the energy stress sensor AMPK has been previously reported to regulate the Hippo signaling pathway through direct phosphorylation of YAP1, but not TAZ, at multiple sites (Mo et al, 2015; Wang et al, 2015), we examine activation of AMPK in DGUOK KO cells. Indeed, there was a marked increase in the levels of active AMPK (pT172-AMPK) and the phosphorylation levels of the AMPK substrate acetyl-CoA carboxylase (pS79-ACC) after DGUOK depletion (Fig 4D). It is interesting to note that DGUOK was also able to activate AMPK and decrease the levels of YAP1 protein in A549 (Fig EV4D), an LKB1 mutant cell line, and the ectopic expression of NDI1 was able to reverse the effect of DGUOK KO, suggesting that the AMPK-YAP1 signaling axis downstream of DGUOK is conserved between LKB1 mutant and wild-type lung cancer. To determine whether the activation of AMPK was responsible for inhibition of YAP1, we used AMPK inhibitor compound C to treat

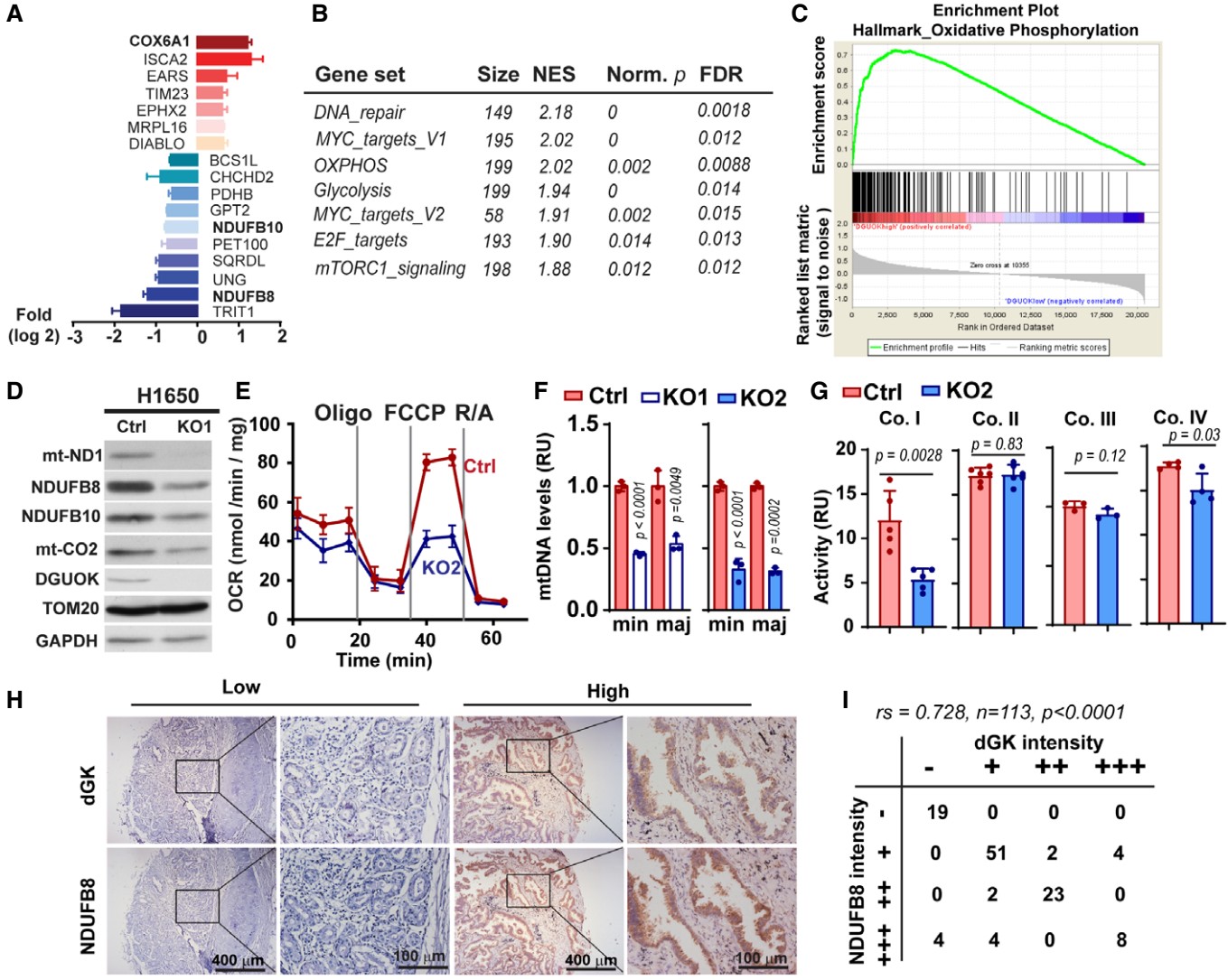

**Figure 3. DGUOK is required for mitochondrial OXPHOS in lung adenocarcinoma cells.**

A   Waterfall plot showing fold changes of 17 mitochondrial proteins differentially regulated in DGUOK KO H1650 cells. Fold changes (log$_2$) are presented as mean ± SEM of results from two independent experiments.

B   Top enriched pathways in *DGUOK* high lung adenocarcinoma (n = 100) compared to *DGUOK* low lung adenocarcinoma (n = 100) based on Gene Set Enrichment Analysis on RNA-sequencing data from the TCGA database. NES, normalized enrichment score, Norm. *p*, normalized *P* value.

C   Enrichment score profile of the oxidative_phosphorylation pathway in GSEA in (B).

D   Western blotting showing the decrease in respiration complex I subunits in DGUOK KO H1650 cells.

E   The effects of DGUOK depletion on mitochondrial oxygen consumption rate in H1650 cells. n = 3 independent replicates per group.

F   The effects of DGUOK KO on mtDNA copies in H1650 cells. Two pairs of primers recognizing minor Arc (minArc) and major Arc (majArc) region of mtDNA were used for the qPCR quantitation.

G   The effect of DGUOK depletion on the activities of respiratory complexes I, II, III, and IV.

H   Representative IHC staining showing the expression of DGUOK and NDUFB8 in lung adenocarcinoma patient specimens.

I   Correlation between DGUOK staining intensity and NDUFB8 staining intensity in a cohort of 113 lung adenocarcinoma patients. *r*s = 0.728, *P* < 0.0001, as determined by Spearman's correlation test.

Data information: Data are shown as mean ± SD. Experiments in (D–F) were repeated with three biological replicates with similar results. n = 3 in (E and F). n = 5 in (G). *P* values in (F and G) were determined by two-tailed, two-sample Student's *t*-test.

Source data are available online for this figure.

control and DGUOK KO cells. Compound C treatment remarkably reduced the levels of pT172-AMPK and pS79-ACC (Fig 4D), confirming the efficacy of the inhibitor treatment. Compound C treatment also restored the protein expression of YAP1 to levels comparable to the control group (Fig 4D). Due to the lack of specific antibodies recognizing AMPK phosphorylated YAP1, we used Phos-tag and compound C to evaluate the role of AMPK in YAP1 phosphorylation. Phos-tag is able to bind to phosphorylated

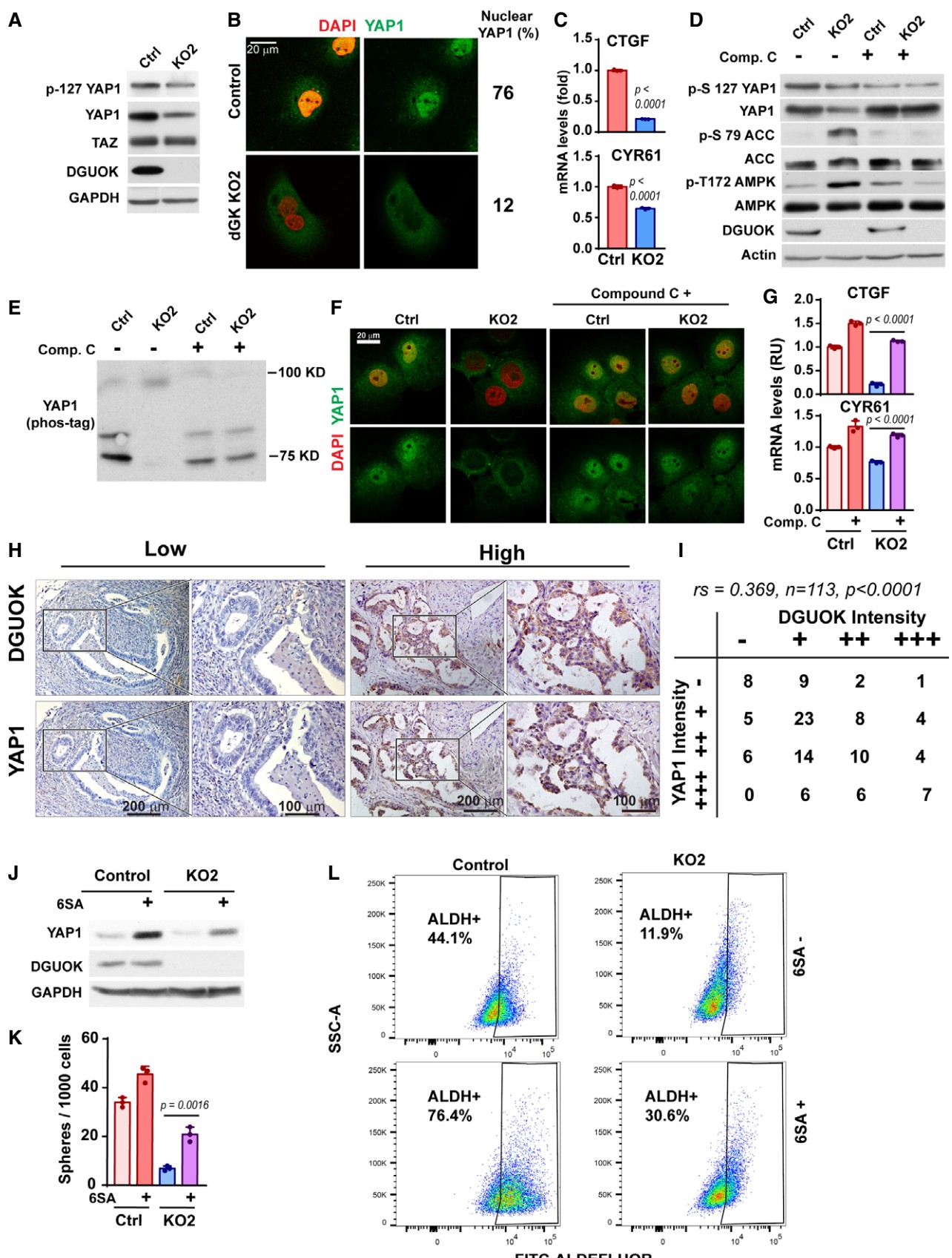

Figure 4.

**Figure 4.  DGUOK depletion inhibits lung adenocarcinoma CSC through an AMPK-YAP1 signaling axis.**

A    The effects of DGUOK depletion on the YAP1 total protein levels and p-S127 YAP1 levels in H1650 cells, as determined by Western blotting.
B, C    The effects of DGUOK depletion in H1650 cells on YAP1 nuclear translocation (B) and the mRNA transcript levels of YAP/TAZ target genes CTGF and CYR61 (C), as determined by immunofluorescence staining and qPCR, respectively.
D    The effects of DGUOK depletion and compound C treatment on the levels of activated AMPK (pT172-AMPK), phosphorylated ACC (pS79-ACC), total YAP1 and pS127-YAP1 in H1650 cells, as determined by Western blotting.
E    The effects of DGUOK depletion and compound C treatment on the phosphorylation status of YAP1, as determined by Phos-tag gel electrophoresis and YAP1 Western blotting.
F, G    The effects of DGUOK depletion and compound C treatment on YAP1 nuclear translocation (F) and the mRNA expression of CTGF, CYR61 (G) in H1650 cells.
H    Representative IHC staining showing the expression of DGUOK and YAP1 in lung adenocarcinoma patient specimens.
I    Correlation between DGUOK staining intensity and NDUFB8 staining intensity in a cohort of 113 lung adenocarcinoma patients. $rs$ = 0.369, $P < 0.0001$, as determined by Spearman's correlation test.
J    Expression levels of YAP1 protein in control and DGUOK KO cells with or without ectopically expressed 6SA-YAP1 mutant, as determined by Western blotting.
K    The effects of ectopic 6SA-YAP1 on lung tumor sphere formation abilities in control and DGUOK KO H1650 cells.
L    The effects of ectopic 6SA-YAP1 in control and DGUOK KO H1650 cells on ALDH$^+$ populations as determined by flow cytometric using ALDEFLUOR assay.

Data information: Data are shown as mean ± SD. $n$ = 3 independent replicates in (C, G and K). $P$ values were determined by two-tailed, two-sample Student's $t$-test. Experiments in (D, E, and J) were repeated with two biological replicates with similar results; experiments in (A–C, F, G, L) were repeated with three biological replicates with similar results.

Source data are available online for this figure.

residues in a protein to cause a mobility shift in SDS–PAGE. In Phos-tag gel, we were able to detect three major YAP1 bands in the control H1650 cells, reflecting different phosphorylation status of YAP1 in these cells (Fig 4E). When DGUOK was KO, all the YAP1 shifted to the slower migrating 95 kDa band, indicating an increase in YAP1 phosphorylation (Fig 4E). Inhibition of AMPK in DGUOK KO cells with compound C robustly reduced the levels of the 95 kDa band while restoring the levels of fast-migrating YAP1, providing further evidence that DGUOK depletion inhibits YAP1 signaling through AMPK-mediated phosphorylation of YAP1 (Fig 4E). Consistent with this notion, compound C treatment was able to restore the levels of nuclear YAP1 and the mRNA expression of the YAP1 target genes CTGF and CYR 61 in DGUOK KO cells (Fig 4F and G).

To determine whether DGUOK may regulate YAP1 expression in lung adenocarcinoma patients, we examine the correlation between DGUOK expression levels and YAP1 levels in the lung adenocarcinoma tissue microarray. As shown in Fig 4H and I, the expression of YAP1 modestly but positively correlated with DGUOK in the lung adenocarcinoma tissue microarray (Spearman's rank $rs$ = 0.369, $P < 0.0001$). Lung adenocarcinoma specimens with high DGUOK expression levels also tend to have increased YAP1 staining (Fig 4H and I), which is consistent with the notion that DGUOK regulates YAP1 in lung cancer patients. Next, we investigated the role of YAP1 in DGUOK-mediated cancer cell stemness by ectopically expressing YAP1-6SA, a constitutively active mutant of YAP1, in control and DGUOK KO cells (Fig 4J). The ectopic expression of YAP1-6SA at least partially restored the lung sphere formation ability and the proportion of ALDH$^+$ CSC in DGUOK KO cells (Figs 4K and L, and EV4E), suggesting that DGUOK depletion inhibits lung cancer cell stemness by inhibiting YAP1 signaling.

## DGUOK regulates CSC self-renewal and YAP1 through mitochondrial OXPHOS

We took two independent approaches to determine whether DGUOK ablation regulates AMPK-YAP1 signaling through mitochondrial OXPHOS. In the first approach, we ectopically express human DGUOK in DGUOK knockout LLC (a murine lung

adenocarcinoma line) cells. In the second approach, we used NDI1, a single subunit NADH: ubiquinone oxidoreductase from yeast, to bypass complex I and allow mitochondria to use NADH as an electron donor for electron transport and oxidative phosphorylation (Maas *et al*, 2010). When expressed in DGUOK KO cells, ectopic DGUOK or NDI1 was able to restore basal, ATP-linked, and maximum capacity OCR (Fig 5A–D). To determine whether the DGUOK-mediated regulation of AMPK-YAP1 signaling was due to inhibition of mitochondrial OXPHOS, we investigated the effects of ectopic DGUOK and NDI1 on AMPK activation and YAP1 expression in DGUOK KO cells. As shown in Fig 5E, the ectopic expression of DGUOK or NDI1 was also able to reduce the levels of pT172-AMPK and pS79-ACC and increase the expression levels of total YAP1. Consistent with the downregulation of active AMPK levels, ectopic DGUOK and NDI1 was also able to reduce the levels of the 95 kDa YAP1 band in Phos-tag gel, indicating a reduction in overall YAP1 phosphorylation levels (Fig 5F). The partial restoration of the mRNA levels of YAP1 target genes CTGF and CYR61 indicates that restoration of mitochondrial OXPHOS by NDI1 was able to rescue YAP1-mediated transcription in these cells (Fig 5G). Taken together, our data suggest that DGUOK depletion regulates AMPK-YAP1 signaling circuit through mitochondrial OXPHOS.

The ectopic expression of DGUOK or NDI1 in DGUOK KO lung cancer cells was able to at least partially restore the proportion of ALDH$^+$ population, CD166$^+$ population, and "side population" when compared to control cells (Figs 6A–C and EV5A–C). Ectopic DGUOK and NDI1 also restore the lung sphere formation in DGUOK KO cells (Fig 6D and E), indicating the restoration of mitochondrial OXOPHOS was able to rescue CSC self-renewal. To further critically evaluate the role of DGUOK and mitochondrial OXPHOS in CSC self-renewal, we used a limiting dilution tumor initiation assay to determine the effects of DGUOK depletion, DGUOK rescue, and NDI1 rescue on cancer cell stemness. Serially diluted LLC cells were injected subcutaneously into syngeneic Albino BL6 mice, and the formation of palpable tumor was determined 26 days after injection. Tumor-negative mice were further tracked for more than 4 months to confirm the inability of these mice to form tumor at the injection sites. As shown in Fig 6F, the tumor initiation probabilities for

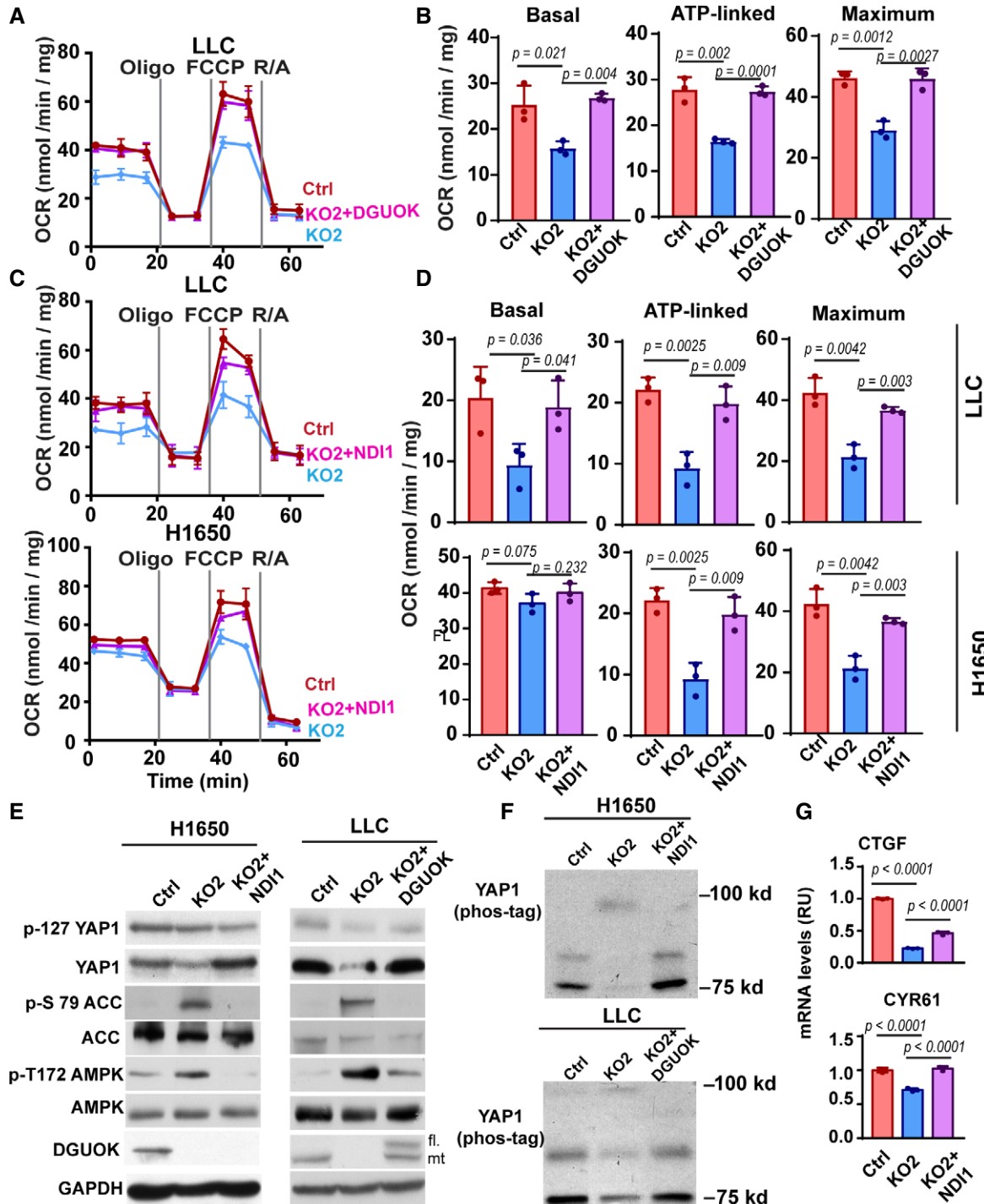

**Figure 5. DGUOK abrogation regulates AMPK-YAP1 signaling through mitochondrial OXPHOS.**

A, B   The effects of ectopic DGUOK on mitochondrial OCR in DGUOK KO LLC, as determined by Mito Stress assay.
C, D   The effects of ectopic yeast NDI1 on mitochondrial OCR in DGUOK KO LLC and H1650 cells.
E, F   Western blotting results showing that the ectopic expression of DGUOK or yeast NDI1 was able to restore total YAP1 levels, inhibit AMPK activation (E), reduce YAP1 phosphorylation (F) in DGUOK KO H1650 cells or LLC cells. In (E), *fl* and *mt* indicate *full-length* (with the mitochondrial targeting signaling peptide) and *mitochondrial* DGUOK (without the signaling peptide), respectively.
G   qPCR results showing that the ectopic NDI1 was able to rescue the expression of CTGF and CYR61 in DGUOK KO H1650 cells.

Data information: Data are shown as mean ± SD. *n* = 3 independent replicates in (A–D and G). *P* values in (B, D, and G) were determined by two-tailed, two-sample Student's *t*-test. Experiments in (E and F) were repeated with two biological replicates with similar results.
Source data are available online for this figure.

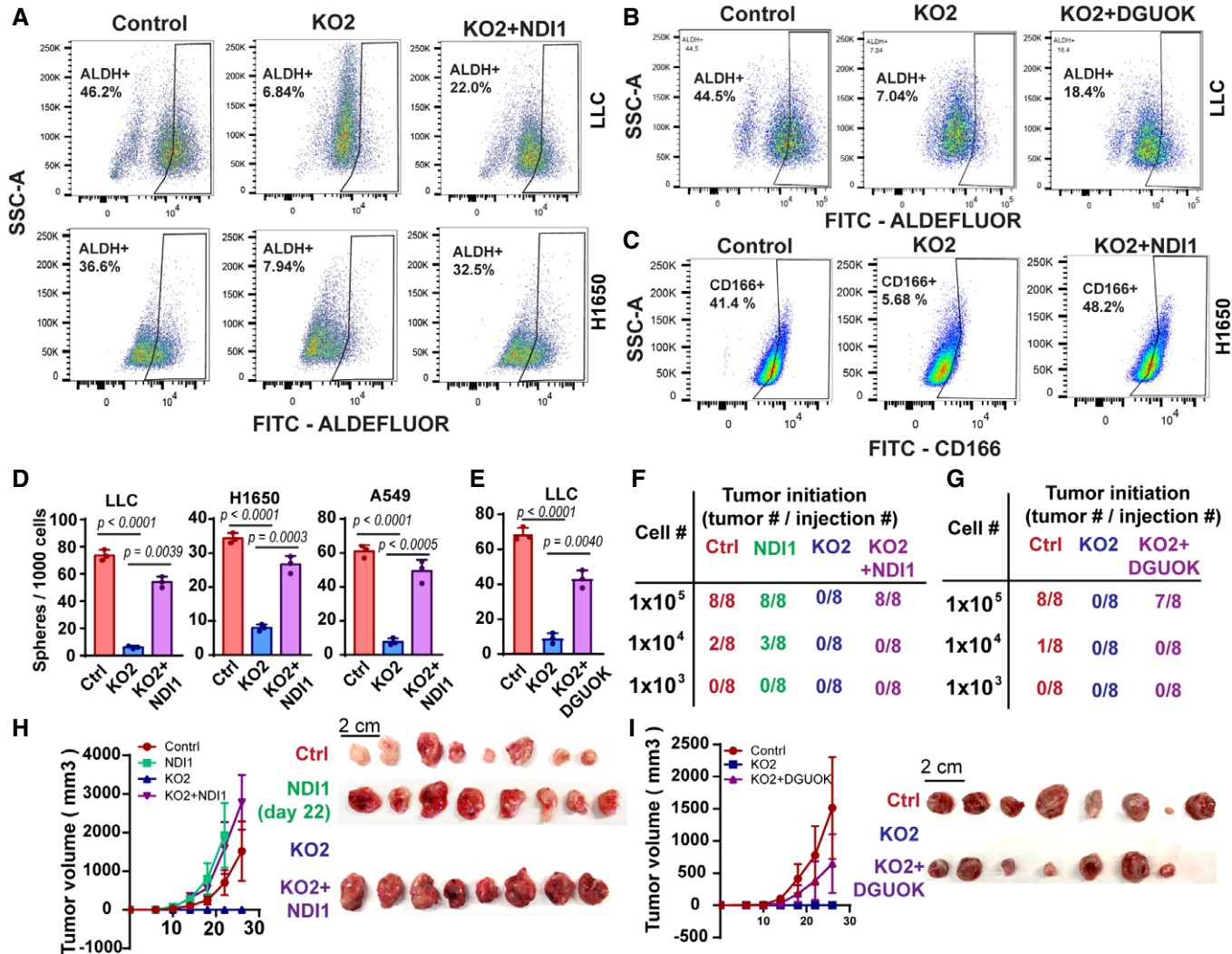

**Figure 6. DGUOK depletion regulates CSC self-renewal by inhibiting mitochondrial OXPHOS.**

A, B The ectopic expression of yeast NDI1 (A) and DGUOK (B) in DGUOK KO H1650 and LLC cells was able to partially restore the levels of ALDH⁺ CSC population, as determined by flow cytometric and ALDEFLUOR assay.

C The ectopic expression NDI1 in GUOK KO H1650 cell was able to restore the CD166⁺ CSC population, as determined by flow cytometric.

D, E The ectopic expression NDI1 (D) and DGUOK (E) was able to restore lung tumor sphere formation in DGUOK KO lung adenocarcinoma cells. Data are shown as mean ± SD (*n* = 3 independent replicates per group).

F, G Limiting dilution tumor initiation assay to determine the effects of DGUOK depletion and NDI1 rescue (F) or DGUOK rescue (G) on CSC self-renewal in LLC cells.

H, I The effects of NDI1 (H) and ectopic DGUOK (I) in DGUOK KO LLC cells on tumor growth (1 × 10⁵ cells). Data are shown as mean ± SD (*n* = 8). Experiment in NDI1 alone group was ended on day 22 due to tumor ulceration. Other groups were ended on day 26. Left panel, tumor growth curves from each group of mice. Right panel, the image of harvested LLC allografts tumors after euthanasia of mice on day 26 (1 × 10⁵ cell groups).

Data information: *P* values in (D and E) were determined by two-tailed, two-sample *t*-test. Experiments in (A and B) were repeated with three biological replicates with similar results.

control LLC cells were 100% (8 out of 8), 25% (2 out of 8), and 0% (0 out of 8) at $1 \times 10^5$, $1 \times 10^4$, and $1 \times 10^3$ cells per injection, respectively. No tumor formation was detected in the DGUOK KO group at any dilution (Fig 6F and G), supporting an essential role for DGUOK in the self-renewal of lung cancer CSC. The ectopic expression of NDI1 in DGUOK KO LLC cells was able to rescue tumor formation at the $1 \times 10^5$ cell dilution (8 out of 8), but not at higher dilutions ($1 \times 10^4$ or $1 \times 10^3$) suggesting at least partial rescue of cancer cell stemness in DGUOK KO cells by restoration of

mitochondrial OXPHOS (Fig 6F). NDI overexpression in control cells modestly increased tumor initiation in $1 \times 10^4$ group (3 out of 8) (Fig 6F). Similarly, ectopic expression of DGUOK in DGUOK KO LLC cells was able to partially restore tumor initiation at $1 \times 10^5$ dilution (7 out of 8) (Fig 6G). Consistent with the tumor initiation experiment, ectopic NDI1 and DGUOK were able to restore tumor growth in DGUOK KO LLC cells (Figs 6H and I, and EV5D and E). Intriguingly, the tumors from the NDI1 OE groups ($1 \times 10^5$ dilution, in either control or DGUOK KO LLC cells) were moderately larger than

the control group (Fig 6H), implicating a role of mitochondrial OXPHOS in tumor growth.

### DGUOK targeting induced tumor regression and YAP1 inhibition

Next, we sought to determine whether the over-reliance of lung CSC on DGUOK could be exploited to inhibit tumor growth and to prevent tumor initiation. There is currently no specific DGUOK inhibitor available; however, several FDA-approved anti-HIV reverse transcriptase inhibitors such as didanosine (DDI) and zidovudine have been shown to inhibit DGUOK and mtDNA homeostasis in cell culture and mouse models (Sun et al, 2014b,c). Interestingly, there is anecdotal evidence that DDI treatment might be responsible for the long-term survival in a HIV-positive small-cell lung cancer patient (Kato et al, 2005). We investigated whether DDI could be repurposed as a DGUOK inhibitor to suppress CSC self-renewal in lung adenocarcinoma. As shown in Fig 7A and B, DDI treatment in H1650 and three patient-derived lung adenocarcinoma lines (PDC027, PDC236, and PDC251) resulted in dramatic reduction in mtDNA copies and the protein levels of DGUOK, confirming earlier observations (Sun et al, 2014b). DDI treatment also robustly decreased the ability of PDC cells to form tumor sphere (Fig 7C and D). The administration DDI to Albino BL6 mice 2 days after the subcutaneous implantation of LLC cells was able to modestly inhibited tumor growth by approximately 50% (Appendix Fig S3C–E).

To determine whether DGUOK targeting could prevent tumor initiation, we used a Tet-On CRISPR/Cas9 system, where doxycycline treatment could efficiently induce the expression of Cas9 and the depletion of DGUOK protein levels in approximately 1 week (Fig 7E). Luciferase-labeled LLC cells expressing Tet-On Cas9 and DGUOK sgRNA were injected into Albino BL6 mice. Two days after injection, mice were randomized into two groups, with one group provided with doxycycline chow to induce the depletion of DGUOK in cancer cells. As shown in Fig 7F, doxycycline-induced DGUOK depletion inhibited 90% of tumor initiation. To evaluate the efficacies of DGUOK targeting in established tumors, LLC allograft expressing Tet-On Cas9 and DGUOK sgRNA was allowed to grow for 12 days until palpable tumors were developed. As control, LLC allografts without DGUOK sgRNA were used. The mice were then randomized into two groups and provided with regular chow or doxycycline chow 12 days postimplantation. As shown in Fig 7G, 7 of the 10 LLC tumor allografts in the DGUOK KO2 group started to regress about 1 week after switching to doxycycline chow (when most tumor measured 50–100 mm$^3$). In contrast, doxycycline chow had no effect on tumor growth in the LLC control group (Fig 7G), suggesting that the tumor regression in the KO group was due to doxycycline-induced depletion of DGUOK. When the experiment ended 2 weeks after the doxycycline chow treatment, most of the regressing tumors had reduced to a size too small for tissue harvesting. The three tumors that did not regress also grew slower than average tumors in the control group (Fig 7G–I). The Western blotting analysis of homogenized tumors demonstrated about 50% decreases in DGUOK protein levels in the three remaining tumors in the doxycycline group, suggesting partial DGUOK depletion in these tumors (Fig 7J). Doxy-induced depletion of DGUOK also reduced the levels of NDUFB8 and YAP1, as determined by Western blotting (Fig 7J). IHC staining of harvested tumor tissues from the LLC model further confirmed the reduction of YAP1 levels from DGUOK

depletion (Fig 7K). Our data suggest that genetic targeting of DGUOK induced tumor regression through inhibition of mitochondrial OXPHOS and YAP1 signaling. Therefore, the dysregulated mitochondrial dNTP metabolism in lung adenocarcinoma CSC could be potentially targeted in lung cancer.

## Discussion

The role of mitochondrial OXPHOS in CSC self-renewal is cancer type-dependent (Loureiro et al, 2017). In CSCs from breast cancer and hepatocellular carcinoma, mitochondrial OXPHOS is suppressed (Dong et al, 2013; Chen et al, 2016), while in lung cancer and pancreatic cancer, mitochondrial OXPHOS is essential for CSC self-renewal (Ye et al, 2011; Sancho et al, 2015). Intriguingly, the mtDNA copies were elevated in lung adenocarcinoma when compared to paired normal lung tissue (Reznik et al, 2016). The dependence on mitochondrial OXPHOS in lung adenocarcinoma could be in part due to the mutations of components of the SWI/SNF tumor suppressor complex (Lissanu Deribe et al, 2018). Here, we provide evidence that DGUOK expression levels are highly elevated in lung adenocarcinoma patients. Depletion of DGUOK in lung adenocarcinoma cells markedly inhibited tumor growth and lung colonization in xenograft and allograft mouse models. Our data further indicate that DGUOK depletion inhibits mitochondrial OXPHOS in lung adenocarcinoma by perturbing mtDNA homeostasis and the biogenesis of mitochondrial respiratory complexes. The inhibition of mitochondrial metabolism severely reduced CSC self-renewal through the AMPK-YAP1 signaling circuit. Importantly, genetic targeting of DGUOK using inducible CRISPR/Cas9 was able to induce robust tumor regression in Lewi's lung carcinoma models.

Although there is currently no specific DGUOK inhibitor available, several reports indicated that FDA-approved anti-HIV reverse transcriptase inhibitors such as DDI and zidovudine could also inhibit DGUOK and TK2 (Sun et al, 2014b,c). Indeed, it was reported that HIV-infected patients receiving DDI treatment displayed 47% decrease in mtDNA copies in liver biopsies (Walker et al, 2004). Intriguingly, there is evidence that DDI treatment might have improved the survival of a small-cell lung cancer patient with HIV infection (Kato et al, 2005). Our data indicate that DDI treatment was able to decrease mtDNA copies and inhibit lung sphere formation in lung adenocarcinoma cell lines and patient-derived cancer cells. Although DDI as an anti-cancer agent in Lewi's lung cancer allograft model was only modestly effective, it is possible that more potent and specific DGUOK inhibitors based on these anti-HIV nucleotide analogs could be developed as therapeutics for lung adenocarcinoma patients.

The loss-of-function mutations of DGUOK gene have been associated with autosomal recessive mitochondrial DNA depletion and/or mitochondrial DNA deletion with diverse clinical phenotypes. In most severe cases, DGUOK mutations may cause fatal infantile hepatocerebral mitochondrial DNA depletion syndrome (Mandel et al, 2001). However, there is also evidence suggesting that loss-of-function DGUOK mutations result in much milder clinical presentations such as limb weakness, myopathy, and portal vein hypertension in adult patients (Ronchi et al, 2012; El-Hattab & Scaglia, 2013; Vilarinho et al, 2016). Intriguingly, Dguok knockout mice appeared to be normal except for female infertility (Sun and Yang, unpublished

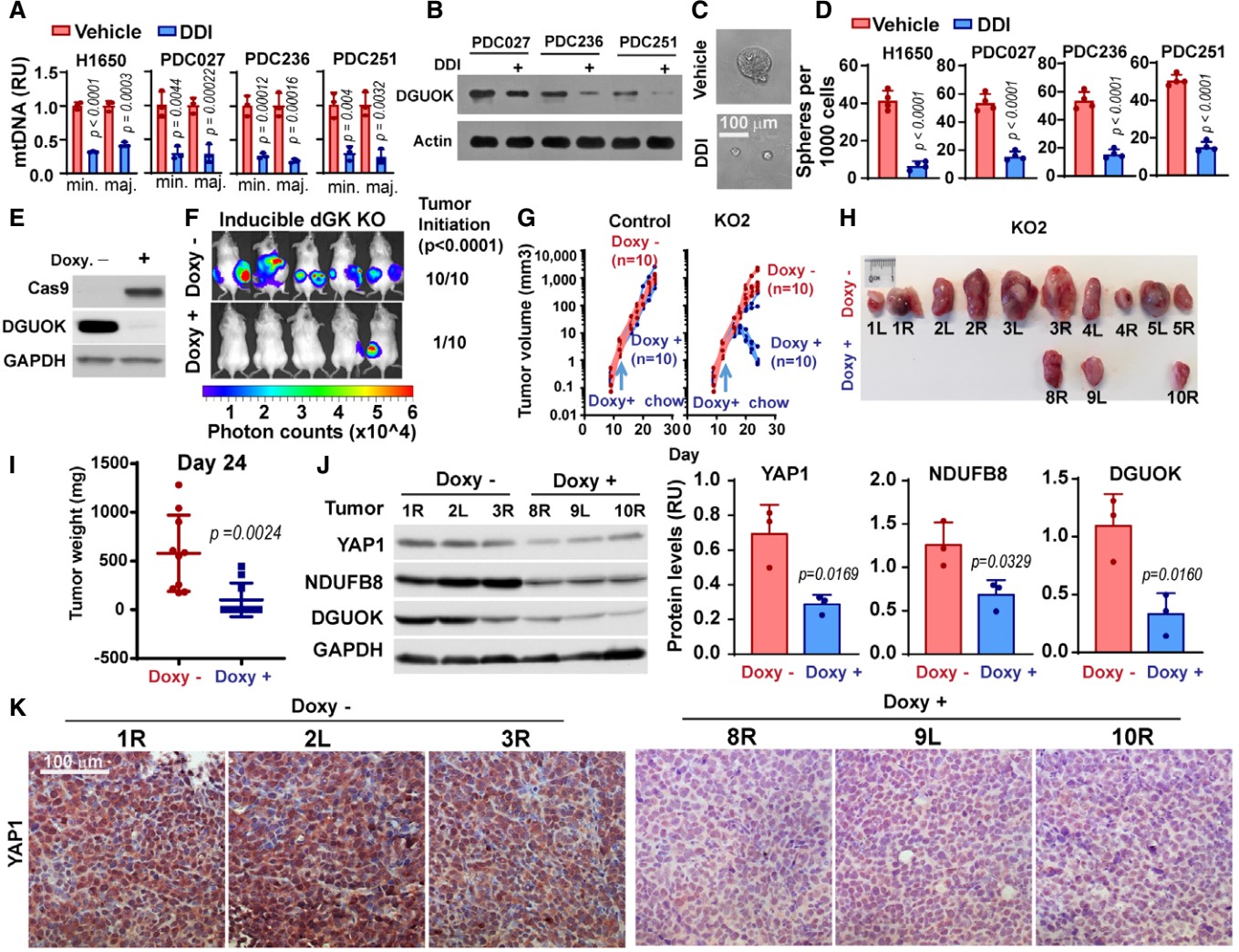

**Figure 7. Genetic targeting of DGUOK inhibits tumor initiation and induces tumor regression.**

A   The effect of DDI treatment on mtDNA copies in H1650 cells and three patient-derived cancer cells from lung adenocarcinoma (PDC027, PDC236, and PDC251). Two pairs of primers recognizing minor Arc (minArc) and major Arc (majArc) region of mtDNA were used for the qPCR quantitation. Data are shown as mean ± SD (n = 3 per group).

B   The effect of DDI treatment on DGUOK protein levels in the three patient-derived cancer cell lines.

C, D   Representative images (C) and quantitation data (D) showing the effects of DDI treatment on lung sphere formation in H1650 and patient-derived lung cancer cells. Data are shown as mean ± SD (n = 4 independent samples per group).

E   Western blotting showing the doxycycline (doxy)-induced expression of Cas9 and the deletion of DGUOK protein expression after 7 days of treatment.

F   $1 \times 10^5$ LLC cells expressing Tet-On Cas9 and DGUOK sgRNA5 (KO2) were inoculated via s.c. into both flanks of Albino BL6 mice. One group of mice (n = 5 mice, 10 inoculations per group) were provided with doxycycline chow 2 days after inoculation. The formation of tumor was monitored by bioluminescence imaging 28 days postinoculation. P < 0.0001, n = 10 mice per group, Fisher's exact test.

G   $1 \times 10^5$ LLC cells stably expressing Tet-On Cas9 and with DGUOK sgRNA (KO2 group) or without sgRNA (control group) were inoculated via s.c. into Albino BL6 mice. Mice were provided with doxycycline chow (doxy+ group), with regular chow as control (doxy− group), 12 days (blue arrow) after inoculation (n = 5 mice, 10 inoculations per group). Tumor growth was measured using a caliper starting from day 9.

H, I   Harvested tumors (H) and quantitation of tumor weight (I) from euthanized KO2 mice from experiment in (G). Data are shown as mean ± SD (n = 5 mice).

J   Western blotting to determine the expression levels of YAP1, NDUFB8, and DGUOK in three tumors each from the doxy− and doxy+ group in (H). Densitometry quantitation of relative protein expression levels (normalized against GAPDH) is shown as bar graphs on the right. Data are shown as mean ± SD (n = 3).

K   Immunohistochemistry staining to detect the expression levels of YAP1 in tumors harvested from doxy− and doxy+ experimental groups.

Data information: P values in (A, D, I, and J) were determined by two-tailed, two-sample Student's t-test.
Source data are available online for this figure.

observations). Our data demonstrated that a 50–70% reduction in mtDNA in DGUOK KO lung cancer cells was sufficient to induced robust tumor regression and inhibition of CSC self-renewal. Clinical

data from HIV-infected patients taking DDI or zidovudine as long-term anti-viral therapies indicate that side-effects from 50% mtDNA level decreases are likely manageable. Therefore, although

**Table 2. Antibodies used in the current study.**

| | Dilution | Source | Cat # |
|---|---|---|---|
| Antibodies for Western blot | | | |
| Human DGUOK antibody | 1:5,000 | Santa Cruz | sc-398101 |
| Murine DGUOK antibody | 1:5,000 | Santa Cruz | sc-376267 |
| MT-COX2 antibody | 1:1,000 | Santa Cruz | sc-514489 |
| Tom20 antibody | 1:5,000 | Santa Cruz | sc-11415 |
| Phospho-AMPKα antibody | 1:1,000 | Cell Signaling | 2535 |
| AMPKα antibody | 1:1,000 | Cell Signaling | 2532 |
| Phospho-Acetyl-CoA Carboxylase antibody | 1:500 | Cell Signaling | 11818 |
| Acetyl-CoA Carboxylase antibody | 1:1,000 | Cell Signaling | 3676 |
| Anti-Rabbit, HRP | 1:5,000 | Cell Signaling | 7074 |
| Anti-Mouse, HRP | 1:5,000 | Cell Signaling | 7076 |
| YAP/TAZ antibody | 1:5,000 | Cell Signaling | 8418 |
| YAP antibody | 1:2,000 | Cell Signaling | 14074 |
| Phospho-YAP antibody | 1:2,000 | Cell Signaling | 4911 |
| NDUFA4 antibody | 1:1,000 | Thermo Fisher | PA5-51021 |
| NDUFA9 antibody | 1:1,000 | Thermo Fisher | 459100 |
| NDUFB8 antibody | 1:1,000 | Thermo Fisher | 459210 |
| MT-ND1 antibody | 1:500 | Thermo Fisher | NBP100939 |
| NDUFB10 antibody | 1:2,000 | Thermo Fisher | PA5-51179 |
| Actin antibody | 1:5,000 | Sigma | A2066 |
| GAPDH antibody | 1:5,000 | Sigma | G8795-100 UL |
| Antibodies for immunofluorescence staining | | | |
| YAP antibody | 1:400 | Cell Signaling | 14074S |
| Alexa 488 goat anti-rabbit | 1:1,000 | Thermo Fisher | A11008 |
| Cytokeratin 20 antibody | 1:200 | Invitrogen | PA522125 |
| Antibodies for IHC | | | |
| DGUOK | 1:200 | Santa Cruz | sc-376256 |
| NDUFB8 | 1:200 | Thermo Fisher | 459210 |
| YAP | 1:100 | Cell Signaling | 14074S |

DGUOK-targeted therapies in lung cancer patients may have potential side-effects on mtDNA homeostasis in the liver, these side-effects are possibly tolerable within a therapeutic window. Our data support that the dysregulation of the mitochondrial dNTP metabolism in lung adenocarcinoma is essential for CSC self-renewal. The reliance of the lung cancer CSC on DGUOK for mitochondrial OXPHOS and self-renewal could be potentially targeted to prevent therapeutic resistance and metastatic recurrence in lung adenocarcinoma patients.

Taken together, our data support that DGUOK is required for CSC self-renewal in lung adenocarcinoma. Genetic or pharmacological targeting of DGUOK in lung adenocarcinoma cells robustly reduced the CSC population and inhibited CSC self-renewal and transdifferentiation. Our data also indicate that DGUOK abrogation had little effect on lung cancer cell survival and proliferation. However, our data are not able to rule out the possibility that DGUOK might also play a role in the larger population of lung cancer cells. Therefore, future investigation to differentiate the role of DGUOK in CSC vs. differentiated cancer cells is warranted.

# Materials and Methods

### Antibodies

Details of antibodies and dilutions are listed in Table 2.

### Cell culture

The lung cancer cell lines (H1650, A549, H292, and LLC) were obtained from the Moffitt Cancer Center Lung Cancer Center of Excellence cell line repository and authenticated using Short Tandem Repeat profiling. All cell lines used were free of microbial (including mycoplasma) contamination. H1650 and H292 were cultured in RPMI medium (HyClone, SH30027.FS) supplemented with 10% fetal bovine serum (Atlanta Biological, S11150) and 1% penicillin–streptomycin (Gibco, 15140163). LLC and A549 were cultured in DMEM medium (HyClone, SH30243.FS) supplemented with 10% fetal bovine serum and 1% penicillin–streptomycin. The

primary lung adenocarcinoma cell lines PDC251, PDC027, and PDC236 were purchased from WuXi Pharmatech Company (WuXi, China). PDC251 and PDC027 cells were cultured in RPMI-1640/ 10% FBS; PDC236 was cultured in DMEM basic medium 10% FBS. All the cells were maintained at 37°C in a humidified 5% $CO_2$ incubator.

### Justification for the choice of cell culture models

The initial experiment to examine the role of DGUOK in lung cancer CSC was performed in H1650 cells and further validated in other lung cancer cell lines. We used LLC cell as a murine lung cancer model to examine the role of DGUOK in lung cancer from a different species. Our guiding principle is to validate our findings in as many cell lines as is technically and financially feasible. LLC and the syngeneic Albino BL6 model were used for limited dilution experiment and inducible DGUOK KO experiment, since this model allows us to evaluate the effects of DGUOK targeting in an immune competent tumor microenvironment. We were not able to perform certain experiments in some cell lines due to technical difficulties. For example, LLC cells were not able to survive the Hoechst red/blue double staining, and therefore, we were not able to evaluate the effect of DGUOK KO on "side population" cells in LLC.

### Inhibitor treatment

For compound C treatment, cells were incubated with medium containing 10 μM compound C (Sigma, P5499) at 37°C for 24 h in a 5% $CO_2$ incubator before being used for Western blot. H1650 cells or PDCs were treated with 100 μM DDI (Ark Pharm, AK-87600) for 7 days in regular cell culture conditions before being used for assays.

### Plasmids

The DGUOK knockout was performed using pLenti CRISPR V2 vector (Addgene_52961) encoding sgRNA targeting human DGUOK, and the sequences are as follows: sg1: ACTTAGAAAGAGGCGGCCCG; sg2: AGGAGGAAACGCCCTCGAGT; sg3: CCTTCGATGGAGAGCCTTC G; sg4: CGGGCCGCCTCTTTCTAAGT; sg5: CCCCGAAGGCTCTCCA TCGA or mouse: sg1: CACGAGCGCGTTCCGCAGCG; sg2: TCGTGGA CGCGCCACACGCC; sg3: TCCACGAGCGCGTTCCGCAG; sg4: GGGG CCGCCCCCGTCGTGCA; sg5: ACGCTCGGAGACGACGCAGA. YAP1 wt (Addgene_42555), YAP1 6SA (Addgene_42562), YAP1 sh1 (Add gene_42540), YAP1 sh2 (Addgene_42541), and NDI1 (Add gene_72876) expression plasmids were obtained from Addgene. The retroviral and lentiviral particles were packaged in HEK293 cells using the PEI transfection method and concentrated as previously described (Yang *et al*, 2012).

### Lung adenocarcinoma patient specimens

A total of 113 sequential lung adenocarcinoma tumor tissues were collected from patients who had received radical surgery at Tianjin Medical University Cancer Institute and Hospital (TJMUCH). Retrospective clinicopathological data of these patients, including age, sex, tumor size, regional lymph node status, TNM stage, pathologic

type, and survival data, were also obtained. These formaldehyde-fixed paraffin-embedded tissues were constructed into tissue microarrays for further IHC analysis. Another set of 82 pairs of lung adenocarcinoma tumor tissues and corresponding adjacent non-tumor tissues were collected and sectioned to analyze the differential expression levels of DGUOK in ADC. The usage of these specimens and the patient information was approved by the Ethics Committee of TJMUCH. Informed consent was obtained from all subjects, and the experiments conformed to the principles set out in the WMA Declaration of Helsinki and the Department of Health and Human Services Belmont Report.

### Immunohistochemistry

Immunohistochemistry staining was used to determine DGUOK, NDUFB8, and YAP1 expression levels in tumor and non-tumor tissues. Paraffin-embedded sections of tissues were deparaffinized and then heated in a pressure pot for 3 min to retrieve the antigens. The slides sections were then incubated with primary antibodies (1:200) overnight at 4°C. Antibody binding was detected using a peroxidase-conjugated secondary antibody at 37°C for 30 min. A DAB Substrate Kit (Vector) was used to perform the chromogenic reaction. The intensity of the staining was evaluated by two independent investigators blinded to the clinicopathological data of patients using the following criteria: 0, negative; 1, low; 2, medium; 3, high. The extent of staining was scored as 0, 0% stained; 1, 1– 25% stained; 2, 26–50% stained; 3, 51–100% stained. Five random fields (20× magnification) were evaluated under a light microscope. The final scores were calculated by multiplying the scores of the intensity with those of the extent and dividing the samples into four grades: 0, negative (−); 1–2, low staining (+); 3–5, medium staining (++); and 6–9, high staining (+++).

### Western blotting

Cells were lysed in SDS-IGEPAL buffer (50 mM Tris, pH 8.0, 150 mM NaCl, 1% IGEPAL (Sigma, I8896), 1% SDS (Fisher Scientific, BP166-500), 1 mM protease inhibitors cocktail (Roche, 04693159001) on ice for 1 min. Cells were scraped from the plate and sonicated briefly three times. Then, lysates were then heated at 95°C for 5 min and centrifuged at 20,000 × *g*, at 4°C for 10 min. To prepare cell lysate for detection of protein phosphorylation, cells were lysed in Triton X-100 buffer (50 mM Tris, pH 7.4, 150 mM NaCl, with 1% Triton X-100, 1 mM EDTA, 1 mM phosphatase inhibitor cocktail (Thermo Fisher, 88667), and 1 mM protease inhibitors cocktail on ice for 15 min. 50 μg proteins were separated by SDS–PAGE and transferred onto polyvinylidene difluoride (PVDF, Millipore, IPVH00010). The membranes were incubated in blocking buffer [5% w/v nonfat dry milk in Tris-buffered saline with 0.05% Tween-20 (TBS-T)] for 30 min at room temperature (RT), and incubated with primary antibodies for 16 h at 4°C, followed by incubation with secondary antibodies for 60 min at RT.

### Phos-tag Western blot

The Phos-tag reagents were purchased from Wako Chemicals (304-93526), and gels containing Phos-tag reagents were prepared

according to the manufacturer's instructions. $Mn^{2+}$-Phos-tag SDS–PAGE was prepared by mixing 1.5 M Tris–HCl (0.4 w/v SDS; pH 8.8), 30% acrylamide/0.8% bisacrylamide, 25 μM Phos-tag, 50 μM $MnCl_2$, 10% APS, and TEMED. Protein extracts were loaded to the Phos-tag gel, separated by electrophoresis, transferred to PVDF membranes, and probed with the specific antibodies.

## Immunofluorescence staining and microscopy

Immunofluorescence staining was carried out as previously described with modifications (Sun *et al*, 2014a). 80,000 cells were seeded onto laminin-coated coverslip in 12-well plate overnight. Cells were then fixed with 4% fresh paraformaldehyde (Sigma, 158127) in phosphate-buffered saline (PBS) at RT for 20 min. Cells were incubated with primary antibodies diluted in antibody dilution buffer (2% BSA 0.1% Triton X-100 in TBS) overnight at 4°C. Cells were then incubated with secondary antibodies for 60 min at RT and then with DAPI (Thermo Fisher, D1306) at RT for 10 min. Extensive washing with TBS was performed between each step. After a final wash with TBS, the coverslips were mounted with Fluoromount (Sigma, F4680).

## Mitochondrial stress test

Oxygen consumption rate was determined with a Seahorse XF96 Extracellular Flux Analyzer (Agilent) following protocols recommended by the manufacturer. 10,000 cells were seeded in each well on XF96 microplates. Cells were maintained in a non-buffered assay medium (Agilent) in a non-$CO_2$ incubator for 30 min before the assay. The XF Cell Mito Stress Test Kit (Agilent, 103015-100) was used for the assay. The baseline recordings were followed by sequential injection of 1 μM oligomycin, 1 μM FCCP, and 1 μM rotenone/20 μM antimycin A.

## Soft-agar assay

Each 12-well plate was coated with 1 ml of bottom agar [RPMI/DMEM containing 10% FBS and 0.75% low melting point agar (Fisher)], and 3,000 cells were suspended in 1 ml of top agar (RPMI/DMEM containing 10% FBS and 0.75% low melting point agar) and plated into each well. 1 ml RPMI/DMEM complete medium was added to the top after the agar solidified. Cells were incubated for 14–21 days, and medium was replaced every 3 days. Colonies were stained with iodonitrotetrazolium chloride (Fisher) overnight.

## Sphere formation assay

1,000 cells were grown in 100 μl per well with DMEM:F12 (1:1) (HyClone, SH30023.01), supplemented with 2% B27 (Invitrogen, 17504044), 20 ng/ml EGF (PeproTech, AF-100-15), 20 ng/ml bFGF (PeproTech, 100-18B), 4 μg/ml heparin (Sigma, H4784), and 0.5% methyl cellulose (Sigma, M0512) to prevent cell aggregation. Single-cell suspensions were cultured for 14–28 days for different cell lines.

## 3D spheroid cultures, differentiation, and staining

The assay was carried out as previously described (Lee *et al*, 2007). Matrigel (Corning, 354277) was thawed at 4°C overnight and used for the coating of glass coverslips. $1 \times 10^4$ lung cancer cells were re-suspended in 200 μl Matrigel and plated onto the pre-coated 24-well plate. After incubating at 37°C for 30 min to allow EHS to gel, 500 μl sphere formation medium containing 10% Matrigel was added. To induce transdifferentiation, RPMI medium supplemented with 10% FBS was added (at day 5 for A549 cell and day 7 for H1650). Spheres were then fixed with 4% fresh paraformaldehyde in PBS at RT for 20 min. Spheres were incubated with anti-Cytokeratin 20 antibody (Invitrogen, PA522125) in antibody dilution buffer (2% BSA 0.1% Triton X-100 in TBS) at 4°C for overnight. Spheres were then incubated with a secondary antibody (Invitrogen, A11012) and phalloidin-488 (Invitrogen, A12381) for 60 min at RT, and then with DAPI at RT for 10 min. Extensive washing with TBS was performed between each step 20 min for five times. After a final wash with TBS, the coverslips were mounted with Fluoromount. The images were taken using Leica SP8 confocal microscope.

## ALDEFLUOR assay

The ALDEFLUOR™ kit (STEMCELL Technologies, 01700) was used to separate the population with the high ALDH enzymatic activity which has been considered as the marker of a lung cancer CSC. The assay follows the protocols suggested by the manufacturer. $1 \times 10^6$ cells were suspended in ALDEFLUOR assay buffer containing ALDH substrate (BAAA) and incubated in a water bath 45 min at 37°C. For co-staining, 10 μl anti-CD166 antibody (Invitrogen, 12-1661-82, 1:200 dilution) was added. For the negative control, each sample aliquot was treated with 50 mM diethylaminobenzaldehyde (DEAB) a specific ALDH inhibitor. The cell flow cytometric sorting gates were established using DEAB-treated cells as negative controls.

## Side population assay

Cells were trypsinized and re-suspended in DMEM:F12 with 2% FBS at $1 \times 10^6$ cells/ml and incubated with 4 μg/ml of Hoechst 33342 (Invitrogen, H1399) at 37°C for 120 min. For negative control, 1 mM fumitremorgin C (FTC) (Sigma, F9054) was added. Cells were then incubated with 2 μg/ml propidium iodide (PI) (Sigma, P4170), before analysis, to label non-viable cells. The Hoechst 33342 dye was excited at 350 nm using UV laser, and its fluorescence was analyzed using 400–500 nm BP filter for blue emission and 640–680 nm BP filter in combination with 655 nm LP filter for red emission. Data were acquired using LSRII.

## RNA isolation and qPCR

$1 \times 10^6$ cells were washed with ice-cold PBS, and total RNA was extracted using the RNeasy kit (Qiagen). 1 μg RNA was used for reverse transcription. cDNA was diluted 1:10 and then used for qPCR with ABI qPCR Master Mix (Thermo Fisher). qPCR primer information can be found in Appendix Table S1.

## mtDNA copy number

The mtDNA was quantified by DNeasy kit (Qiagen). $1 \times 10^6$ cells were used for total DNA extraction according to the manufacturer's instructions with the inclusion of RNAse A treatment to generate

RNA-free genomic DNA. Quantitative PCR was performed using DNA extracted from both whole-cell extracts and cytosolic fractions using human or mouse nuclear DNA primers and mtDNA primers listed in Appendix Table S2.

### Inducible CRISPR/Cas9 system

The inducible CRISPR/Cas9 knockout was performed as we recently reported (Lin *et al*, 2019). LLC cells were infected with pLenti-CMV-tight-Cas9-Hygro and pLenti-CMV-rtTA3-Blast (w756-1) (Addgene) and selected with hygromycin and blasticidin. Single clones were induced with 1 μg/ml doxycycline for 3 days and stained with anti-Flag antibody to examine the expression levels of Tet-On Flag-Cas9. Fifteen clones with high Cas9 expression were combined, infected with pLenti-Guide-Puro (Addgene), and then selected with puromycin. The cells were then treated with 1 μg/ml doxycycline for 7 days, and the expression of Flag Cas9 and deletion of DGUOK were examined by Western blotting.

### Quantification of cellular dNTP

The assay was carried out as previously described (Wilson *et al*, 2011). $1 \times 10^6$ cells were plated in a 10-cm tissue culture dishes and cultured for 48 h. The adherent cells were trypsinized and re-suspended gently in ice-cold PBS. The cells were centrifuged at $300 \times g$, 4°C for 5 min, and the cell pellets were vortexed vigorously in 500 μl ice-cold 60% methanol for 1 min. The suspension was then heated at 95°C for 3 min and sonicated. The extracts were centrifuged at $16,000 \times g$ for 5 min at 4°C to remove cell debris, precipitated protein, and DNA. The supernatant was evaporated using centrifugal vacuum at 70°C and reconstituted in 25 μl nuclease-free water.

Reaction mixtures contained primer, probe, and template at an equimolar final concentration of 0.4 mM. Standard dNTPs (Fisher Scientific, BP2564-1) were included for the standard curve. DreamTaq DNA Polymerase (Thermo Scientific, EP0703) was added at 1 U/reaction to a final volume of 25 μl. Thermal profiling and fluorescence detection were performed using the Quant Studio 3 Real-Time PCR System (Thermo Scientific). Raw fluorescence spectra for 6-FAM were measured every 3 min.

### Mitochondrial respiratory complex activity assay

The mitochondrial complex activity assay kits (Abcam ab109721 for complex I, Abcam ab109908 for complex II, BioVision K520-100 for complex III, Abcam ab109909 for complex IV) were used to quantitatively measure the activity of mitochondrial respiratory complexes, respectively. The assays follow the protocols suggested by the manufacturer. For mitochondrial complex III activity, the mitochondria were isolated by using a mitochondrial isolation kit (BioVision K288-50) following the manufacturer's protocol.

### CD166 assay

$1 \times 10^6$ cells were incubated with Alexa Fluor 488-conjugated CD166 antibody (Invitrogen, MA5-23565, 1:200 dilution) in 100 μl PBS/10% FBS for 30 min at room temperature. Cells were washed three times with ice-cold PBS and then re-suspended in 500 μl ice-cold PBS/10% FBS before being used for flow cytometric analysis. The cell flow and PI were used to gate the dead cells.

### Cell death assay

The FITC Annexin V/Dead Cell Apoptosis Kit (Thermo Fisher, V13242) was used to test the cell death under adherent or non-adherent conditions following the manufacturer's protocols. Briefly, $1 \times 10^6$ cells were suspended in 100 μl 1× annexin-binding buffer containing FITC-annexin V (1:20 dilution) and the PI (1:100 dilution) working solution. Cells were incubated at room temperature for 15 min in dark. After the incubation, 400 μl of 1× annexin-binding buffer was added, and the samples were kept on ice for flow cytometric analysis. The cell flow cytometric sorting gates were established using the cells stained with annexin V only or PI only as the negative controls.

### Animal experiments

All animal experiments were performed according to protocols approved by IACUCs at the Moffitt Cancer Center or Penn State College of Medicine. Seven-week-old female nude mice were purchased from Charles River and housed in a pathogen-free room with a 12-h light/dark cycle. For H1650 lung orthotopic experiments, $1 \times 10^6$ luciferase-labeled H1650 cells were re-suspended in 50 μl 0.25 mg/ml Matrigel (Corning) with PBS buffer, and then orthotopically injected into the left lung of 7-week-old female nude mice, as previously described (Onn *et al*, 2003). One-milliliter tuberculin syringes (Becton Dickinson) with 30-gauge hypodermic needles were used to inject the cell inoculum percutaneously into the left lateral thorax, at the lateral dorsal axillary line, _1.5 cm above the lower rib line just below the inferior border of the scapula. The needle was quickly advanced 5–7 mm into the thorax and was quickly removed after the injection of cell suspension. After tumor injection, the mouse was turned to the left lateral decubitus position. Animals were observed for 45–60 min until fully recovered. Noninvasive bioluminescent imaging and analysis were performed as described previously using Xenogen IVIS 200 (Yang *et al*, 2009, 2012; Sun *et al*, 2014a). For H292 lung colonization experiments, $5 \times 10^6$ luciferase-labeled H292 cells were re-suspended in 300 μl RPMI medium (serum-free) and inoculated into 7-week-old female nude mice via tail vein. Noninvasive bioluminescent imaging and analysis were performed as described previously using Xenogen IVIS 200.

For tumor initiation experiments, LLC cells ($1 \times 10^5$, $1 \times 10^4$ or $1 \times 10^3$ cells) were suspended in 100 μl PBS and injected subcutaneously into 5-week-old female Albino BL6 mice. The formation of palpable tumor was monitored every day.

For DDI (Ark Pharm, AK-87600) treatment, 24 h after subcutaneous injection of LLC cells ($1 \times 105$, both flanks), mice were administered with 20 mg/kg DDI via drinking water (*ad libitum*). Drinking water was changed daily for 26 days. After inoculation, mice were randomized to control or treatment groups. The total number of animals in each treatment group was 5.

For doxycycline-inducible knockout DGUOK in mouse xenograft, 5-week-old Albino BL6 female mice were subcutaneously injected (in both flanks) with $1 \times 10^5$ LLC cells stably expressing Tet-On Cas9 and DGUOK sgRNA. The mice were randomly assigned to two

groups (five mice each). One group was provided with doxycycline chow (BioServ, S3888, 200 mg/kg) on day 12 after inoculation, while the other group was continued with regular chow.

## Quantitative proteomics

To compare protein expression levels in control and DGUOK KO cells, chemical labeling using tandem mass tags (TMT) was combined with liquid chromatography–tandem mass spectrometry discovery proteomics. Samples were prepared in biological duplicate. Protein lysates were made using aqueous denaturing buffer containing 20 mM HEPES and 8 M urea, prior to reduction of disulfides, alkylation of cysteines, and tryptic digestion (Worthington). The resulting peptides are extracted using C-18 Sep-Pak cartridges, dried, and re-suspended in 100 mM aqueous triethylammonium bromide (TEAB) for TMT labeling. Each channel is analyzed with LC-MS/MS for quality control (i.e., tag incorporation and reporter ion verification). Then, samples are mixed using equal amounts of total protein and fractionated with basic pH-reversed phase liquid chromatography; each peptide fraction is analyzed in duplicate with LC-MS/MS (RSLCnano and QExactive Plus, Thermo). Data are analyzed with MaxQuant (Cox & Mann, 2008), which identifies each peptide using sequence-specific ions and quantifies relative expression level in each sample using the TMT reporter ions. After iterative rank-order normalization to correct for differences in loading, significant differences were determined as proteins that exceeded two times the standard deviation away from the mean $\log_2$ ratio between the two samples (Welsh *et al*, 2013).

## Quantification and statistical analysis

All data are shown as mean values ± standard deviation (SD) unless indicated otherwise. Sample numbers of data obtained from animal experiments refer to the number of individual mice, as specified in the figure legends. Experiments on cultured cells were performed using at least three independent biological replicates. Log-rank tests were used for survival analysis. Categorical data were analyzed using Fisher's exact test or chi-square test. Spearman's rank-order correlation was used to determine the correlation between IHC staining intensities for different antigens. Comparison of two independent groups was performed with unpaired two-tailed Student's *t*-test. No samples, data points, or animals were excluded from the analysis. Statistical analysis was performed using GraphPad Prism 7 or SPSS software.

# Data availability

The mass spectrometry proteomics data have been deposited to the ProteomeXchange Consortium via the PRIDE partner repository with the dataset identifier PXD014753 and https://doi.org/10.6019/pxd 014753.

**Expanded View** for this article is available online.

## Acknowledgements

This work is supported by grants from the National Cancer Institute (R01CA175741, R01CA233844), Elsa U. Pardee Foundation, and Penn State Cancer Institute Developmental Funds Award to S. Yang. S. Lin is supported by a DOD LCRP Concept Award (W81XWH-18-1-0283). Proteomics is supported in part by the National Cancer Institute through Moffitt's Cancer Center Support Grant (P30-CA076292) and the Moffitt Foundation. We would like to thank Dr. Srikumar Chellappan for help with the lung sphere formation assay.

## Author contributions

SY conceived this project. SL performed most of the lung cancer cell culture-based experiments and xenograft experiments with assistance from JS, XW, EM, and JKa. SY and SL wrote the manuscript with assistance from PKS, CH, JH, and JKo. CH and JH performed all the IHC staining experiments and experiments involved patient-derived lung cancer cells. CH performed the GSEA using TCGA sequencing data. OB and MDT performed the orthotopic lung injection. BF and JKo performed the proteomics and analyzed the results. SY, SL, PKS, CH, JH, and JKo revised the manuscript.

## Conflict of interest

The authors declare that they have no conflict of interest.

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
