## [Review Process File · EMBO Molecular Medicine]

The mitochondrial deoxyguanosine kinase is required for cancer cell stemness in lung adenocarcinoma

Shengchen Lin, Chongbiao Huang, Jianwei Sun, Oana Boltt, Xiuchao Wang, Eric Martine, Jiaxin Kang, Matthew D. Taylor, Bin Fang, Pankaj K. Singh, John Koomen, Jihui Hao, Shengyu Yang

Review timeline:

Submission date:	6 May 2019
Editorial Decision:	22 May 2019
Revision received:	19 August 2019
Editorial Decision:	10 September 2019
Revision received:	17 September 2019
Accepted:	20 September 2019

Editor: Lise Roth

Transaction Report:

1st Editorial Decision

22 May 2019

Thank you for the submission of your manuscript to EMBO Molecular Medicine. We have now received feedback from the three reviewers who agreed to evaluate your manuscript. As you will see from the reports below, the referees acknowledge the interest of the study. However, they also raise substantial concerns on your work, which should be convincingly addressed in a major revision of the present manuscript.

Addressing the reviewers' concerns in full will be necessary for further considering the manuscript in our journal, and acceptance of the manuscript will entail a second round of review. EMBO Molecular Medicine encourages a single round of revision only and therefore, acceptance or rejection of the manuscript will depend on the completeness of your responses included in the next, final version of the manuscript. For this reason, and to save you from any frustrations in the end, I would strongly advise against returning an incomplete revision.

Please note that it is EMBO Molecular Medicine policy to allow only a single round of revision and that, as acceptance or rejection of the manuscript will depend on another round of review, your responses should be as complete as possible.

I look forward to receiving your revised manuscript.

***** Reviewer's comments *****

Referee #1 (Remarks for Author):

This manuscript by Lin and Huang et al describes the role of mitochondrial deoxynucleoside triphosphate (dNTP) pool maintenance by deoxyguanosine kinase (DGUOK) in initiation and

maintenance of lung adenocarcinoma. The authors have provided robust evidence for its link with mitochondrial oxidative phosphorylation (OXPHOS) and AMPK mediated YAP signaling. In the manuscript, the authors have also explored intersection of this mitochondrial metabolism with cancer stem-like cells (CSC), however additional experiments will be helpful to confirm their findings. In addition, experiments evaluating the effect of DGUOK inhibition on cytosolic dNTP pool will also increase the robustness of mitochondrial specificity of these findings. Overall, this study is particularly relevant, considering increasing interest in metabolic reprogramming of cancer cells as a therapeutic target, and recognition of OXPHOS as an essential metabolic process in some cancer types and subpopulations. Of note, Molina et al recently provided preclinical evidence for activity of IACS-010759, a small-molecule inhibitor of complex I of the mitochondrial electron transport chain, targeting OXPHOS pathway and the rationale for its testing in phase I clinical trials (Molina et al, *Nat Med* 2018, PMID 29892070). This manuscript is suitable for a full length report in the journal.

MAJOR CRITICISMS/ SUGGESTIONS:

1. The authors have used ALDH and CD166 as markers for cancer stem cells in NSCLC, and provided appropriate references in the manuscript. However, both markers individually may not necessarily capture a homogeneous subpopulation of cancer stem cells. For example, Zhang et al in their 2017 paper (referenced in this manuscript) expanded their CSC markers to CD166+CD49^{hi}CD104-Lin- for human specimens, compared to solely CD166+ cells in their 2012 paper (referenced in this manuscript). It would be interesting to see how much of an overlap is between ALDH high and CD166 high cells.

(a) We recommend simultaneous staining for ALDH and CD166 to identify double positive population to strengthen the authors' claim of identifying cancer stem-like cells.

(b) Evaluation for co-expression of other NSCLC stem cell markers, for example CD49^{high} would further enhance the robustness of these findings. (Ghosh et al, *Am J Respir Cell Mol Biol* 2011, PMID 21131442).

2. Have the authors checked the effect of DGUOK KO on cytosolic dNTP pool in these experiments? Mitochondrial DNA replication effect has been described earlier to disturb cellular dNTP pool, which can have an impact on cell survival and proliferation (Nikkanen et al, *Cell Metabolism* 2016, PMID 26924217; Gandhi et al, *Nucleosides Nucleotides Nucleic Acids* 2011, PMID 21774628).

3. AMPK studies were done in H1650 (EGFR mutant cell line). (a) Can the authors show results for AMPK studies for the other cell line A549 (KRAS & LKB-1 mutant cell line) used in the experiment? LKB1 is a well-known upstream activator/ phosphorylator of AMPK, and the studies in A549 will help clarify if these proposed mechanistic findings are broadly applicable (Lin et al, *Cell Metabolism* 2018, PMID 29153408; Steinberg et al, *Nat Rev Drug Discov* 2019, PMID 30867601).

4. Regarding Table 1, the authors have performed 'multivariable analysis' and not 'multivariate analysis'. There is only one dependent variable (overall survival), and multiple independent variables (tumor size, DGUOK level etc.). The differences between the two analyses can be found in the reference, Hidalgo B et al, *Am J Public Health* 2013, PMID 23153131. Additionally,

(a) The variables in the multivariable analysis, tumor size and TNM staging are not truly independent from each other (tumor size is included in TNM staging). The authors should assess for multicollinearity by examining tolerance and the variance inflation factor (VIF).

(b) Both age and smoking status should be included as independent variables in the multivariable analysis (despite non-significance in univariate analysis). References regarding association of smoking with survival in non-small cell lung cancer are: Lee et al, *Thorac Cancer* 2014, PMID 26766971; Kim et al, *Oncotarget* 2017 PMID 29190984.

(c) Do the authors have information on the presence of actionable ('druggable') driver mutations in these patients? Treatment with targeted therapies matched with the driver mutation leads to improved overall survival in lung adenocarcinoma.

MINOR CRITICISMS/ SUGGESTIONS:

1. In Fig. 3F & 7A showing effect of DGUOK KO on mtDNA copies, can authors please explain what is meant by min and maj in figure legends?
2. Regarding fig. 4I, the positive correlation between DGUOK and YAP1 on lung adenocarcinoma specimens is only weak, with value of coefficient R_s being 0.369. The weakness of the correlation should be mentioned in the text.

Referee #2 (Comments on Novelty/Model System for Author):

No issues

Referee #2 (Remarks for Author):

Our understanding of mitochondrial deoxynucleoside salvage pathway in cancer is very limited and I found the study an interesting read.

Major comments:

- The link to 'stem cells' is quite tenuous and could be clarified. Few of the experiments look specifically at putative 'stem cells' and so results likely reflect that DGUOK knockout has fitness effects in the bulk tumour population. Flow cytometry experiments that look at potential stem cell markers CD166 and ALDH are unconvincing. How were gates determined for these experiments? There are often not clear populations of negative and positive cells as the authors interpretation of these data implies: in treatment conditions the median expression of a single populations seem to change. Gates often bisect these single populations and so the definition of 'stem cells' seems arbitrary. Given that effects can be seen in western blots on bulk populations, isn't it likely that the DGUOK knockout affects a large proportion of cells rather than a subpopulation?
- Cell lines are used inconsistently - H1650 in CSC assay and orthotopic/subcutaneous in vivo models, H292 in tail vein model, A549 in CSC assays. Patient cultures are only introduced in Fig 7. The authors do not need to repeat all experiments in all cell lines but the choice of model should be justified.
- For primary cell experiments, what are the expression levels of DGUOK in these cells? Does CRISPR KO have any phenotype? It would be reassuring to do some basic confirmatory work in primary cells since they are available.
- In Figure 1J the authors refer to right lobe tumours as metastases. What is the proposed mechanism of spread from left to right lobe (could it be a technical problem with injection, needle track etc.?). Do the authors see any other metastases? Why are only excised lungs shown for this experiment (Fig 1G clearly shows bioluminescence can be seen in the lungs of whole mice in this model)?
- Is this role for DGUOK common to all cells or specific to lung adenocarcinoma? In particular, very simple experiments looking at normal lung epithelial cells or fibroblasts would further the authors case to pursue DGUOK inhibition in the clinic.
- I find the experiment in Figure 4K and 4L difficult to reconcile with the authors other data. YAP1-6SA is a constitutively inactive YAP1 - the S94A mutation prevents TEAD interaction. How can an inactive mutant restore lung sphere formation etc. after DGUOK depletion? The authors hypothesis suggests that this mutant should not alter the DGUOK knockout phenotype? A constitutively active mutant should have the effect seen here?

Minor comments:

- There is an 's' missing from cells in the manuscript title.
- The cancer statistics in the introduction are almost 25 years old - these are published annually so should be updated.
- Data are described as 'remarkable' 9 times in the manuscript which is distracting.
- In places the DGUOK CRISPR-Cas9 cell lines are referred to as 'depleted' but they are knockout cells?
- How do the authors justify the selection of NDUFB8 (vs. other Complex I subunits) for further analysis in the tissue microarray?
- Statistics are performed on some qPCR experiments (Fig 5) but not others (Fig 4).
- GAPDH in Fig 4A is massively overexposed, making it uninterpretable.

- Experimental repeat information is provided in legends but it is not clear what was repeated, e.g. in Figure 4D, were blots repeated in independent KO lines? Same cells on different days? The same lysates?
- Labels are missing from Fig 4J.
- In Figure 4K, expressing YAP1-6SA increases sphere formation in controls, so the increase seen in the knockout cells might not represent 'partial rescue'?
- Why is the side scatter profile of KO + YAP1-6SA cells so different to the three other conditions in Fig 4L?
- NDI treatment rescues the DGUOK-KO phenotype but does not 'rescue' AMPK-YAP signalling as the authors suggest in the introduction. Rather it prevents AMPK-mediated phosphorylation of YAP, preventing its nuclear exclusion.
- PDAC is commonly used as an abbreviation for pancreatic ductal adenocarcinoma so the authors might avoid using it in this context.
- The story is complex - a schematic model showing the authors proposed explanation for their data would be very useful, either in a final figure or in graphical abstract form.

Referee #3 (Comments on Novelty/Model System for Author):

The major claim of this study is the novel role for mitochondrial dNTP metabolism in lung cancer tumor growth and progression, and implicate that the mitochondrial deoxynucleotide salvage pathway could be potentially targeted to prevent therapy resistance and metastasis. These findings are novel and have not been reported in previous literatures. The mechanisms of how DGUOK regulate cancer cells stemness were well elucidated with numerous techniques, which are convincing and outstanding. These knowledge are important in CSC metabolism and development of new strategy of therapy. I think this paper is interesting for audience in the fields of oncology and biology.

Referee #3 (Remarks for Author):

This is a well-structured study with merits of robust techniques and abundant data presentation. Authors hypothesized mitochondrial deoxynucleoside salvage pathway and DGUOK as potential targets in lung cancer stem cells (CSC), which are slow-cycling and responsible for metastasis and recurrence. Authors proved that DGUOK overexpression correlates with survival of lung cancer patients in meta-analysis datasets and lung adenocarcinoma progression in tissue microarray. DGUOK depletion inhibited tumor growth and metastasis in vivo and inhibited CSC population and sphere formation. Moreover, authors demonstrated that oxidative phosphorylation is highly enriched in DGUOK-high lung adenocarcinoma patients via GSEA analysis of lung adenocarcinoma RNAseq data. In mechanism, authors demonstrated that DGUOK is required for the biogenesis of respiratory Complex I and mitochondrial OXPHOS, which in turn regulates CSC self-renewal through AMPK-YAP1 signaling. Although there is no available inhibitor of DGUOK, authors used didanosine to inhibit sphere formation in H1650 and patient-derived lung adenocarcinoma cells, and demonstrated that genetic targeting DGUOK could induce tumor regression.

Major comments:

1. Authors demonstrated that DGUOK is required for cancer cells stemness, and quoted that the ability of tumor sphere formation under non-adherent condition with serum-free medium is considered a surrogate assay for the self-renewal abilities of cancer cell stemness. However, the trans-differentiation phenotype is also important in self-renewal ability of CSC. Changes of trans-differentiation in the model of DGUOK depletion should be investigated to emphasize the role of DGUOK in CSC stemness.
2. Authors demonstrated that DGUOK depletion inhibits the ability of tumor growth and metastasis of cancer cells (Fig. II & 1J). The development of metastasis indicates epithelial-mesenchymal transition (EMT), one of the important phenotypes of CSC. Assays and markers of EMT in the condition of DGUOK depletion were suggested.
3. In Fig. 5E, the quality of blots of p-S 79 ACC and ACC in H1650 seems not good. Can you provide the better one? What does "fl mt" (beside blot of DGUOK) mean? It should be addressed in figure legend.

Minor comments:

Some problems about figures:

1. Why the KO cells in Fig. 2A & 2B were different?
2. In Fig. 3D, did you have the data of KO2 cells? Or KO1 should be KO2? Data of Figure 1~3 were constructed of Control/KO1/KO2, and the data after Figure 3 were constructed of Control vs. KO2.
3. In Fig. 3G & 3I, and Fig. S2E: Does "dGK" mean DGUOK? If it does, I guess you could replace it with "DGUOK" (if the length of words is acceptable) to unify the label.
4. Fig. 4J should label the group of "Ctrl vs. KO2"

Some words should be corrected in several places:

1. Page 4: 19th line: ...cells were "orthotopically" injected
2. Page 5: first line of the last paragraph: we "used" mito-stress; 6th line: we "investigated" the effect...
3. Page 6: 2nd line of 2nd paragraph: we "used" immunohistochemistry; 4th line: NDUFB8 "colocalized" with ...
4. Page 7: 9th line of last paragraph: we "investigated" the effects of ectopic
5. Page 8: 11th line of the 2nd paragraph: 4 months to "confirm" the inability...; Does "PDAC" cells in the 12th line of the last paragraph mean three PDC cells from patients?
6. Page 24: (L) should be added after "...in paired para-tumor lung tissues" in legend of 1B; "Scale" bars in 1C
7. Page 25: 9th line of Fig. 4 legend: F and G, "the"; 10th line: "mRNA" should be added before "expression of CTGF..."; 17th line: ALDEFLUOR "assay"... "substraten=3 independent..." should be revised.
8. 10th line of legend of Supplementary Fig S1: F, negative control.... in Figure "2A" (not Figure 1I).

1st Revision - authors' response

19 August 2019

Referee #1 (Remarks for Author):

This manuscript by Lin and Huang et al describes the role of mitochondrial deoxynucleoside triphosphate (dNTP) pool maintenance by deoxyguanosine kinase (DGUOK) in initiation and maintenance of lung adenocarcinoma. The authors have provided robust evidence for its link with mitochondrial oxidative phosphorylation (OXPHOS) and AMPK mediated YAP signaling. In the manuscript, the authors have also explored intersection of this mitochondrial metabolism with cancer stem-like cells (CSC), however additional experiments will be helpful to confirm their findings. In addition, experiments evaluating the effect of DGUOK inhibition on cytosolic dNTP pool will also increase the robustness of mitochondrial specificity of these findings. Overall, this study is particularly relevant, considering increasing interest in metabolic reprogramming of cancer cells as a therapeutic target, and recognition of OXPHOS as an essential metabolic process in some cancer types and subpopulations. Of note, Molina et al recently provided preclinical evidence for activity of IACS-010759, a small-molecule inhibitor of complex I of the mitochondrial electron transport chain, targeting OXPHOS pathway and the rationale for its testing in phase I clinical trials (Molina et al, Nat Med 2018, PMID 29892070). This manuscript is suitable for a full length report in the journal.

Response: We would like to thank this reviewer for expressing his/her interest and enthusiasm in our findings.

MAJOR CRITICISMS/ SUGGESTIONS:

1. The authors have used ALDH and CD166 as markers for cancer stem cells in NSCLC, and provided appropriate references in the manuscript. However, both markers individually may not necessarily capture a homogeneous subpopulation of cancer stem cells. For example, Zhang et al in their 2017 paper (referenced in this manuscript) expanded their CSC markers to CD166+CD49fhiCD104-Lin- for human specimens, compared to solely CD166+ cells in their 2012 paper (referenced in this manuscript). It would be interesting to see how much of an overlap is between ALDH high and CD166 high cells.
 - (a) We recommend simultaneous staining for ALDH and CD166 to identify double positive population to strengthen the authors' claim of identifying cancer stem-like cells.

Response: As suggested by this reviewer, we performed ALDEFLUOR/CD166 double staining to identify the double positive population in H1650 cells and the data are shown in Fig. EV 2B. In the double staining experiment majority of the ALDH+ (25.8% of the 32.8%) are also positive for CD166 staining, suggesting that the ALDH+ CSC population and CD166+ population largely overlapped. Importantly, DGUOK KO markedly decreased the double positive cells to 1.21%, which is consistent with the crucial role of DGUOK in lung cancer CSC.

(b) Evaluation for co-expression of other NSCLC stem cell markers, for example CD49^{high} would further enhance the robustness of these findings. (Ghosh et al, Am J Respir Cell Mol Biol 2011, PMID 21131442).

Response: Our data indicated that CD49f is highly expressed in lung cancer cell lines (100% of the cells were positive (Fig. EV 2C and EV 2D)). Although DGUOK KO reduced the staining intensity of CD49f (Fig. EV 2D), it didn't change the proportion of CD49f+ cells (Fig. EV 2C). Since we were not able to identify a CD49f- population, we didn't pursue the double staining experiment.

2. Have the authors checked the effect of DGUOK KO on cytosolic dNTP pool in these experiments? Mitochondrial DNA replication effect has been described earlier to disturb cellular dNTP pool, which can have an impact on cell survival and proliferation (Nikkanen et al, Cell Metabolism 2016, PMID 26924217; Gandhi et al, Nucleosides Nucleotides Nucleic Acids 2011, PMID 21774628).

Response: We would like to thank this reviewer for raising an interesting point. It was reported in the literature that mtDNA loss might either increase or decrease the cellular dNTP pool. We did examine the effects of DGUOK KO on cellular dNTP levels and our data showed decrease levels of all four dNTPs in DGUOK KO lung cancer cells (Fig. EV 3F).

3. AMPK studies were done in H1650 (EGFR mutant cell line). (a) Can the authors show results for AMPK studies for the other cell line A549 (KRAS & LKB-1 mutant cell line) used in the experiment? LKB1 is a well-known upstream activator/ phosphorylator of AMPK, and the studies in A549 will help clarify if these proposed mechanistic findings are broadly applicable (Lin et al, Cell Metabolism 2018, PMID 29153408; Steinberg et al, Nat Rev Drug Discov 2019, PMID 30867601).

Response: Our new experiments in the LKB1 mutant A549 cells indicated that DGUOK KO could also activate AMPK and decrease YAP1 protein levels (Fig. EV 4D), which could be restored by the ectopic expression of NDI1. These data indicate that the AMPK-YAP1 signaling could be broadly applicable in even LKB1-mutant lung cancer, potentially through CaMKK β , another upstream activator of AMPK.

4. Regarding Table 1, the authors have performed 'multivariable analysis' and not 'multivariate analysis'. There is only one dependent variable (overall survival), and multiple independent variables (tumor size, DGUOK level etc.). The differences between the two analyses can be found in the reference, Hidalgo B et al, Am J Public Health 2013, PMID 23153131. Additionally, (a) The variables in the multivariable analysis, tumor size and TNM staging are not truly independent from each other (tumor size is included in TNM staging). The authors should assess for multicollinearity by examining tolerance and the variance inflation factor (VIF). (b) Both age and smoking status should be included as independent variables in the multivariable analysis (despite non-significance in univariate analysis). References regarding association of smoking with survival in non-small cell lung cancer are: Lee et al, Thorac Cancer 2014, PMID 26766971; Kim et al, Oncotarget 2017 PMID 29190984. (c) Do the authors have information on the presence of actionable ('druggable') driver mutations in these patients? Treatment with targeted therapies matched with the driver mutation leads to improved overall survival in lung adenocarcinoma.

Response: Yes, the analysis method used in Table 1 was actually multivariable analysis instead of multivariate analysis. We have now corrected this mistake in the text. As suggested by this reviewer, we performed a multicollinearity diagnostics in SPSS to evaluate the independence between tumor size and TNM staging. The tolerance and the variance inflation factor was 0.134 and 7.463, respectively, suggesting multicollinearity was present between the two factors. Therefore, we excluded TNM staging in multivariable analysis. In addition, both age and smoking status was included in the multivariable analysis of the revised Table 1. We do not have any information on the

presence of actionable driver mutations in these patients, for gene sequencing was not performed in these specimens.

MINOR CRITICISMS/ SUGGESTIONS:

1. In Fig. 3F & 7A showing effect of DGUOK KO on mtDNA copies, can authors please explain what is meant by min and maj in figure legends?

Response: Following notation has been added in the figure legends for clarity:

Two pairs of primers recognizing minor Arc (minArc) and major Arc (majArc) region of mtDNA were used for the qPCR quantitation.

2. Regarding fig. 4I, the positive correlation between DGUOK and YAP1 on lung adenocarcinoma specimens is only weak, with value of coefficient Rs being 0.369. The weakness of the correlation should be mentioned in the text.

Response: We have indicated in the revised manuscript that the correlation was modest but positive.

Referee #2 (Comments on Novelty/Model System for Author):

No issues

Referee #2 (Remarks for Author):

Our understanding of mitochondrial deoxynucleoside salvage pathway in cancer is very limited and I found the study an interesting read.

Response: We would like to thank this reviewer for finding our manuscript interesting.

Major comments:

• *The link to 'stem cells' is quite tenuous and could be clarified. Few of the experiments look specifically at putative 'stem cells' and so results likely reflect that DGUOK knockout has fitness effects in the bulk tumour population. Flow cytometry experiments that look at potential stem cell markers CD166 and ALDH are unconvincing. How were gates determined for these experiments? There are often not clear populations of negative and positive cells as the authors interpretation of these data implies: in treatment conditions the median expression of a single populations seem to change. Gates often bisect these single populations and so the definition of 'stem cells' seems arbitrary. Given that effects can be seen in western blots on bulk populations, isn't it likely that the DGUOK knockout affects a large proportion of cells rather than a subpopulation?*

Response: We would like to thank this reviewer for raising an important question and we addressed his/her concern as following:

1. Gates in the ALDEFLOUR experiments were determined using DEAB, an inhibitor for ALDH, as shown in Appendix Fig. S2A. Negative control in the presence of DEAB for each ALDEFLOUR experiments were provided in Fig. EV 2A, EV 4E, EV 5A. Gates for CD166 staining experiments were determined using unstained cells, as shown in Appendix Fig. S2B.
2. As noted by this reviewer, in our ALDEFLOUR and CD166 experiments gates often bisect a single populations (with the exception of ALDEFLOUR staining in LLC cells, cf. Fig. 2A). This phenomenon is consistent with other CSC assays using these two markers in the literature (e.g. PMID: 28819374 and PMID: 28819374). Therefore we used another CSC assay that identified a "side population" CSC that were able to exclude Hoechst dye through ABCG2 (cf. PMID: 17510412). As shown in Fig. EV 2E and Fig. EV 5C, Appendix Fig. 2C, Hoechst staining was able to identified a distinct "side population" CSC that disappeared in the presence of ABCG2 inhibitor FTC. Importantly, DGUOK KO robustly reduced the proportion of "side population" (from 47.5% to 10.4% in H1650, 20.9% to 0.51% in A549), which could be restored by the ectopic expression of yeast NDI1(Fig. EV5C).

3. We have included the following paragraph in the discussion section to acknowledge the possibility that DGUOK might also be critical for the larger population of lung cancer cells. Taken together, our data supported that DGUOK is required for CSC self-renewal in lung adenocarcinoma. Genetic or pharmacological targeting of DGUOK in lung adenocarcinoma cells robustly reduced the CSC population and inhibited CSC self-renewal and transdifferentiation. Our data also indicated that DGUOK abrogation had little effect on lung cancer cell survival and proliferation. However, our data are not able to rule out the possibility that DGUOK might also play a role in the larger population of lung cancer cells. Therefore, future investigation to differentiate the role of DGUOK in CSC versus differentiated cancer cells is warranted.

• *Cell lines are used inconsistently - H1650 in CSC assay and orthotopic/subcutaneous in vivo models, H292 in tail vein model, A549 in CSC assays. Patient cultures are only introduced in Fig 7. The authors do not need to repeat all experiments in all cell lines but the choice of model should be justified.*

Response: This point is well taken. We have now included a **Justification for the choice of cell culture models** in the method section as following:

The initial experiment to examine the role of DGUOK in lung cancer CSC was performed in H1650 cells and further validated in other lung cancer cell lines. We used LLC cell as a murine lung cancer model to examine the role of DGUOK in lung cancer from a different species. Our guiding principle is to validate our findings in as many cell lines as is technically and financially feasible. H1650 cell was used in the orthotopic metastasis experiment but not for tail-vein injection experiment, since H1650 was not able to colonize the lung through tail-vein injection. LLC and the syngeneic Albino BL6 model was used for limited dilution experiment and inducible DGUOK KO experiment, since this model allow us to evaluate the effects of DGUOK targeting in an immune competent tumor microenvironment.

• *For primary cell experiments, what are the expression levels of DGUOK in these cells? Does CRISPR KO have any phenotype? It would be reassuring to do some basic confirmatory work in primary cells since they are available.*

Response: This point is well taken. We have now evaluated the relative expression levels of DGUOK in patient-derived cancer cells (in comparison to H1650 and A549, Fig. EV 2F) and the effect of DGUOK KO on tumor sphere formation, YAP1 levels and NDUFB8 levels in these cells (Fig. EV 2G, EV, 4B). Our new experiments suggested that DGUOK is also essential for Complex I biogenesis and cancer cell stemness in primary lung cancer cells.

• *In Figure 1J the authors refer to right lobe tumours as metastases. What is the proposed mechanism of spread from left to right lobe (could it be a technical problem with injection, needle track etc.?). Do the authors see any other metastases? Why are only excised lungs shown for this experiment (Fig 1G clearly shows bioluminescence can be seen in the lungs of whole mice in this model)?*

Response:

1. The metastasis of lung cancer cells from one side of the lung to the contralateral lung is very common, which happened in 22% to 50% of metastatic lung cancer patients (Hsu et al., 2017; Popper, 2016). Lung cancer cells might spread to the contralateral side by shedding from primary tumor into the pleural effusion and seeding the contralateral lung from pleura effusion. Alternatively, metastatic cells might also spread to the contralateral lung through blood circulation: lung cancer cells → bronchial capillaries → bronchial vein → Vena cava → the right atrium → the right ventricle → pulmonary artery → the contralateral lung.
2. In this experiment we didn't observe metastasis to other organs.
3. The orthotopic lung cancer metastasis experiment was performed following a previously published protocol (PMID: 14654533) and explained in more details in Appendix Fig. S1. The lung cancer cells were directly injected into the left lobe of the lung and a 4 mm needle adaptor was used to ensure that the needle would not penetrate into the right side of the

lung. In a pilot experiment, 5 mice were euthanized immediately after injection and the localization of injected lung cancer cells were determined using BLI imaging. As shown in Appendix Fig. S1D-II, lung cancer cells were only detected in the left lungs.

4. Representative whole mouse BLI imaging were shown in Fig. EV. 1F and the quantitation of whole mouse BLI signals from 10 mice were shown in Fig. EV 1G (n=10 in each group).

• Is this role for DGUOK common to all cells or specific to lung adenocarcinoma? In particular, very simple experiments looking at normal lung epithelial cells or fibroblasts would further the authors case to pursue DGUOK inhibition in the clinic.

Response: This point is well taken. We have now examined the effect of DGUOK KO on complex I protein levels in NIH3T3 fibroblast cells. As shown in Fig. EV 3A, DGUOK KO had no effect on the expression of mtND1, NDUFB8 or YAP1, indicating that the role of DGUOK KO might be specific to lung adenocarcinoma. We would also like to mention that patients with DGUOK loss-of-function mutations has no defect in the lung, suggesting that DGUOK is dispensable in normal lung epithelial cells.

• I find the experiment in Figure 4K and 4L difficult to reconcile with the authors other data. YAP1-6SA is a constitutively inactive YAP1 - the S94A mutation prevents TEAD interaction. How can an inactive mutant restore lung sphere formation etc. after DGUOK depletion? The authors hypothesis suggests that this mutant should not alter the DGUOK knockout phenotype? A constitutively active mutant should have the effect seen here?

Response: The YAP1 -6SA mutant was obtained from Addgene (#42562, reported in PMID: 23245941) and is a constitutively active mutant (instead of dominant negative). The mutated residues are S61A, S109A, S127A, S128A, S131A, S163A, S164A, S381A and there is no S94A mutations.

Minor comments:

• There is an 's' missing from cells in the manuscript title.

Response: The typo has been corrected. Thank you.

• The cancer statistics in the introduction are almost 25 years old - these are published annually so should be updated.

Response: A new cancer statistics publication from 2015 has been cited.

• Data are described as 'remarkable' 9 times in the manuscript which is distracting.

Response: The text has been edited to reduce the repetitive use of “remarkable”

• In places the DGUOK CRISPR-Cas9 cell lines are referred to as 'depleted' but they are knockout cells?

Response: “depleted” has now been replaced with “KO” in the manuscript.

• How do the authors justify the selection of NDUFB8 (vs. other Complex I subunits) for further analysis in the tissue microarray?

Response: NDUFB8 was selected because: a) this is the most down-regulated complex I protein in DGUOK KO cells; and b) there was selective NDUFB8 antibody suitable for IHC staining.

• Statistics are performed on some qPCR experiments (Fig 5) but not others (Fig 4).

Response: We have now ensure that statistical analysis are performed for all the data when appropriate.

• GAPDH in Fig 4A is massively overexposed, making it uninterpretable.

Response: A lighter exposure of GAPDH blot has been provided.

• Experimental repeat information is provided in legends but it is not clear what was repeated, e.g.

in Figure 4D, were blots repeated in independent KO lines? Same cells on different days? The same lysates?

Response: The figure legends has been modified to improve clarity in experimental repeat information.

• Labels are missing from Fig 4J.

Response: This has been corrected. Thank you!

• In Figure 4K, expressing YAP1-6SA increases sphere formation in controls, so the increase seen in the knockout cells might not represent 'partial rescue'?

Response: We considered this a “partial rescue” because the fold increase from ectopically expressed YAP1-6SA was significantly higher in KO2 cells than in control cells.

• Why is the side scatter profile of KO + YAP1-6SA cells so different to the three other conditions in Fig 4L?

Response: Thank you for pointing this out. We have repeated the experiment again and the side scattering of KO + YAP1-6SA group was similar to other groups (Fig.

• NDI treatment rescues the DGUOK-KO phenotype but does not 'rescue' AMPK-YAP signaling as the authors suggest in the introduction. Rather it prevents AMPK-mediated phosphorylation of YAP, preventing its nuclear exclusion.

Response: The sentence in the Abstract has been modified as following:

The restoration of mitochondrial OXPHOS in DGUOK KO lung cancer cells using NDI1, a single subunit yeast NADH : ubiquinone oxidoreductase, was able to **prevents AMPK-mediated phosphorylation of YAP and to rescue CSC stemness.**

• PDAC is commonly used as an abbreviation for pancreatic ductal adenocarcinoma so the authors might avoid using it in this context.

Response: PDAC has been replaced with “PDC” to avoid confusion.

• The story is complex - a schematic model showing the authors proposed explanation for their data would be very useful, either in a final figure or in graphical abstract form.

Response: A graphical abstract has been submitted in this revision (in synopsis). Thanks for the suggestion.

Referee #3 (Comments on Novelty/Model System for Author):

The major claim of this study is the novel role for mitochondrial dNTP metabolism in lung cancer tumor growth and progression, and implicate that the mitochondrial deoxynucleotide salvage pathway could be potentially targeted to prevent therapy resistance and metastasis. These findings are novel and have not been reported in previous literatures. The mechanisms of how DGUOK regulate cancer cells stemness were well elucidated with numerous techniques, which are convincing and outstanding. These knowledge are important in CSC metabolism and development of new strategy of therapy. I think this paper is interesting for audience in the fields of oncology and biology.

Referee #3 (Remarks for Author):

This is a well-structured study with merits of robust techniques and abundant data presentation. Authors hypothesized mitochondrial deoxynucleoside salvage pathway and DGUOK as potential targets in lung cancer stem cells (CSC), which are slow-cycling and responsible for metastasis and recurrence. Authors proved that DGUOK overexpression correlates with survival of lung cancer patients in meta-analysis datasets and lung adenocarcinoma progression in tissue microarray. DGUOK depletion inhibited tumor growth and metastasis in vivo and inhibited CSC population and sphere formation. Moreover, authors demonstrated that oxidative phosphorylation is highly enriched in DGUOK-high lung adenocarcinoma patients via GSEA analysis of lung adenocarcinoma RNAseq data. In mechanism, authors demonstrated that DGUOK is required for the biogenesis of respiratory Complex I and mitochondrial OXPHOS, which in turn regulates CSC

self-renewal through AMPK-YAP1 signaling. Although there is no available inhibitor of DGUOK, authors used didanosine to inhibit sphere formation in H1650 and patient-derived lung adenocarcinoma cells, and demonstrated that genetic targeting DGUOK could induce tumor regression.

Response: We would like to thank this reviewer for finding our findings novel, and our data “convincing and outstanding”.

Major comments:

1. Authors demonstrated that DGUOK is required for cancer cells stemness, and quoted that the ability of tumor sphere formation under non-adherent condition with serum-free medium is considered a surrogate assay for the self-renewal abilities of cancer cell stemness. However, the trans-differentiation phenotype is also important in self-renewal ability of CSC. Changes of trans-differentiation in the model of DGUOK depletion should be investigated to emphasize the role of DGUOK in CSC stemness.

Response: We thank the reviewer for raising this interesting point. We have now evaluated the effects of DGUOK KO on serum-induced transdifferentiation of lung sphere. As shown in Fig. 2G and 2H, when embedded in Matrigel control H1650 and A549 cells were able to form tumor spheres with low or undetectable levels of CK20 in 5 days. The formation of tumor sphere was inhibited in DGUOK KO cells. 10% FBS added at day 5 was able to induce robust CK20 expression in 2-5 days in control lung cancer cells, but not in DGUOK KO cells, suggesting that DGUOK is crucial for the transdifferentiation of lung CSC.

2. Authors demonstrated that DGUOK depletion inhibits the ability of tumor growth and metastasis of cancer cells (Fig. 1I & 1J). The development of metastasis indicates epithelial-mesenchymal transition (EMT), one of the important phenotypes of CSC. Assays and markers of EMT in the condition of DGUOK depletion were suggested.

Response: As suggested by this reviewer, we examined the effect of DGUOK KO on cancer cell migration, invasion and the expression of several EMT markers, including E-cadherin, vimentin, Slug and β -catenin. Although DGUOK KO dramatically inhibited lung cancer cell migration and invasion (Appendix Fig. S1A-S1B), there was no obvious change in the expression of EMT markers (Appendix Fig. S1C).

3. In Fig. 5E, the quality of blots of p-S 79 ACC and ACC in H1650 seems not good. Can you provide the better one? What does "fl mt" (beside blot of DGUOK) mean? It should be addressed in figure legend.

Response: We were able to re-run the p-S 79 ACC and ACC blot and replaced the data in Fig. 5E. *fl* and *mt* indicate *full length* (with the mitochondrial targeting signaling peptide) and *mitochondrial* DGUOK (without the signaling peptide), respectively. This information has now been included in the figure legend.

Minor comments:

Some problems about figures:

1. Why the KO cells in Fig. 2A & 2B were different?

Response: The CD166 antibody only recognized human CD166 but not murine CD166, therefore we are not able to evaluate the effect of DGUOK KO on CD166 population in LLC cells. Since we wanted to validate each experiment in at least 2 cell lines whenever feasible, we used A549 instead of LLC cells.

2. In Fig. 3D, did you have the data of KO2 cells? Or KO1 should be KO2? Data of Figure 1~3 were constructed of Control/KO1/KO2, and the data after Figure 3 were constructed of Control vs. KO2.

Response: KO2 data were shown in Fig. EV 3A.

3. In Fig. 3G & 3I, and Fig. S2E: Does "dGK" mean DGUOK? If it does, I guess you could replace it with "DGUOK" (if the length of words is acceptable) to unify the label.

Response: Yes, we have now replaced it with DGUOK.

4. Fig. 4J should label the group of "Ctrl vs. KO2"

Response: Yes, we have now corrected the mistake. Thanks for pointing this out.

Some words should be corrected in several places:

1. Page 4: 19th line: ...cells were "orthotopically" injected

2. Page 5: first line of the last paragraph: we "used" mito-stress; 6th line: we "investigated" the effect...

3. Page 6: 2nd line of 2nd paragraph: we "used" immunohistochemistry; 4th line: NDUFB8 "colocalized" with ...

4. Page 7: 9th line of last paragraph: we "investigated" the effects of ectopic

5. Page 8: 11th line of the 2nd paragraph: 4 months to "confirm" the inability...; Does "PDAC" cells in the 12th line of the last paragraph mean three PDC cells from patients?

6. Page 24: (L) should be added after "...in paired para-tumor lung tissues" in legend of 1B; "Scale" bars in 1C

7. Page 25: 9th line of Fig. 4 legend: F and G, "the"; 10th line: "mRNA" should be added before "expression of CTGF..."; 17th line: ALDEFLUOR "assay"... "substraten=3 independent..." should be revised.

8. 10th line of legend of Supplementary Fig S1: F, negative control.... in Figure "2A" (not Figure 1I).

Response: The above typos/mistakes have all been corrected. Thank you!

2nd Editorial Decision

10 September 2019

Thank you for the submission of your revised manuscript to EMBO Molecular Medicine. We have now received the referees' reports, and as you will see they are supportive of publication of your study pending minor revisions. I am therefore pleased to inform you that we will be able to accept your manuscript once you have addressed the comments from the three referees.

I look forward to reading a new revised version of your manuscript as soon as possible.

***** Reviewer's comments *****

Referee #1 (Comments on Novelty/Model System for Author):

Cell lines and mouse models they used were adequate, no ethical issues

Referee #1 (Remarks for Author):

I appreciate the efforts by the authors to revise the manuscript, and am satisfied with their responses. Minor Comments:

The point of 50% reduction of mitochondrial DNA copies in liver in HIV patients taking Didanosine has been repeated in consecutive paragraphs in the discussion section. This should be consolidated for brevity and to avoid redundancy.

Referee #2 (Remarks for Author):

The manuscript has improved in light of the reviewers comments and the authors have broadly addressed my concerns outlined in response to the first submission. The addition of additional experiments around cancer stem cells in particular is welcome, although I would still suggest to that while the data support an effect of DGUOK KO on "cancer stem cell" markers/phenotypes they do not support the idea that DGUOK KO's principal anti-tumour role is in a subset of tumour cells, as signalling effects are seen in the bulk population.

Minor comments:

1) I remain uncomfortable with the flow data as presented - with gates that bisect populations and then classification of 'CSC' populations as % - would it be better presented as histograms with MFI to show the population shift instead?

- 2) The manuscript still contains many typos/inconsistencies - the journal might consider editing by a native English speaker before publication.
- 3) 'Remarkably' still appears 7 times in the manuscript.
- 4) The importance of CK20 expression in the new spheroid assays (Fig. 2G/2H) was not clear to me.
- 5) I could not find the graphical abstract in my submission files so cannot comment on it.
- 6) In figures, the first bar has bidirectional error bars but then subsequent bars only have error bars above.
- 7) Annexin V is misspelt in Fig. 2G.

Referee #3 (Comments on Novelty/Model System for Author):

The major claim of this study is the novel role for mitochondrial dNTP metabolism in lung cancer tumor growth and progression, and implicate that the mitochondrial deoxynucleotide salvage pathway could be potentially targeted to prevent therapy resistance and metastasis. These findings are novel and have not been reported in previous literatures. The mechanisms of how DGUOK regulate cancer cells stemness were well elucidated with numerous techniques, which are convincing and outstanding. These knowledge are important in CSC metabolism and development of new strategy of therapy. I think this paper is interesting for audience in the fields of oncology and biology.

Referee #3 (Remarks for Author):

This revised manuscript provided more abundant data including double staining of ALDH and CD166, dNTP level, AMPK-YAP1 signaling, trans-differentiation assay in lung CSCs and DGUOK KO cells. Table 1 was also revised after re-evaluating independency of variables. Authors have explained the choice of cell culture model and checked DGUOK expression in patient-derived cancer cells and lung fibroblasts. These data are convincing and helpful to clarify the role of DGUOK in the self-renewal of lung CSCs.

I only have some minor suggestion about typo errors: (Personally, I don't think spelling errors should appear in high class journal.)

1. Page 4, 19th line: ...cells were "orthotopically" injected
2. Page 24: "Scale" bars in 1C

2nd Revision - authors' response

17 September 2019

***** **Reviewer's comments** *****

Referee #1 (Comments on Novelty/Model System for Author):

Cell lines and mouse models they used were adequate, no ethical issues

Referee #1 (Remarks for Author):

I appreciate the efforts by the authors to revise the manuscript, and am satisfied with their responses.

Minor Comments:

The point of 50% reduction of mitochondrial DNA copies in liver in HIV patients taking Didanosine has been repeated in consecutive paragraphs in the discussion section. This should be consolidated for brevity and to avoid redundancy.

Response: The discussion section has been revised to consolidate the information, as suggested by reviewer 1.

Referee #2 (Remarks for Author):

The manuscript has improved in light of the reviewers comments and the authors have broadly

addressed my concerns outlined in response to the first submission. The addition of additional experiments around cancer stem cells in particular is welcome, although I would still suggest to that while the data support an effect of DGUOK KO on "cancer stem cell" markers/phenotypes they do not support the idea that DGUOK KO's principal anti-tumour role is in a subset of tumour cells, as signalling effects are seen in the bulk population.

Response: We agree with this reviewer that our data were not able to rule out the possibility that DGUOK might also be critical for non-CSC and we had included the following statement to discuss this possibility:

Genetic or pharmacological targeting of DGUOK in lung adenocarcinoma cells robustly reduced the CSC population and inhibited CSC self-renewal and transdifferentiation. Our data also indicated that DGUOK abrogation had little effect on lung cancer cell survival and proliferation. However, our data are not able to rule out the possibility that DGUOK might also play a role in the larger population of lung cancer cells. Therefore, future investigation to differentiate the role of DGUOK in CSC versus differentiated cancer cells is warranted.

Minor comments:

1) I remain uncomfortable with the flow data as presented - with gates that bisect populations and then classification of 'CSC' populations as % - would it be better presented as histograms with MFI to show the population shift instead?

Response: To address this minor concern from reviewer 2, we change our reference to "ALDH+ CSC population" or "CD166+ CSC population" to merely "ALDH+ population" and "CD166+ population". Although we appreciate the suggestion to use histograms and MFI to show the population shift, we have determined that such data presentation will not be consistent with the intent of these assays.

2) The manuscript still contains many typos/inconsistencies - the journal might consider editing by a native English speaker before publication.

Response: We have had native speaker co-authors of this manuscript to carefully proof-read the revised manuscript and to correct any mistakes/typos.

3) 'Remarkably' still appears 7 times in the manuscript.

Response: We have reduced the use of "remarkably" to 3 times in this revision.

4) The importance of CK20 expression in the new spheroid assays (Fig. 2G/2H) was not clear to me.

Response: The CK20 staining was part of the transdifferentiation experiment suggested by reviewer 3 in the previous round of revision.

5) I could not find the graphical abstract in my submission files so cannot comment on it.

Response: The graphical abstract has now been uploaded as an individual file.

6) In figures, the first bar has bidirectional error bars but then subsequent bars only have error bars above.

Response: The bidirectional error bars in all the bar graphs has been revised to error bars above the column.

7) Annexin V is misspelt in Fig. 2G.

Response: This typo (and other typos) has been corrected.

Referee #3 (Comments on Novelty/Model System for Author):

The major claim of this study is the novel role for mitochondrial dNTP metabolism in lung cancer tumor growth and progression, and implicate that the mitochondrial deoxynucleotide salvage pathway could be potentially targeted to prevent therapy resistance and metastasis. These findings are novel and have not been reported in previous literatures. The mechanisms of how DGUOK regulate cancer cells stemness were well elucidated with numerous techniques, which are

convincing and outstanding. These knowledge are important in CSC metabolism and development of new strategy of therapy. I think this paper is interesting for audience in the fields of oncology and biology.

Referee #3 (Remarks for Author):

This revised manuscript provided more abundant data including double staining of ALDH and CD166, dNTP level, AMPK-YAP1 signaling, trans-differentiation assay in lung CSCs and DGUOK KO cells. Table 1 was also revised after re-evaluating independency of variables. Authors have explained the choice of cell culture model and checked DGUOK expression in patient-derived cancer cells and lung fibroblasts. These data are convincing and helpful to clarify the role of DGUOK in the self-renewal of lung CSCs.

I only have some minor suggestion about typo errors: (Personally, I don't think spelling errors should appear in high class journal.)

1. Page 4, 19th line: ...cells were "orthotopically" injected

2. Page 24: "Scale" bars in 1C

Response: These 2 typos (and other typos) have been corrected.

Corresponding Author Name: Shengyu Yang

Journal Submitted to: EMBO Mol. Med.

Manuscript Number: EMM-2019-10849